# Optimal or Greedy Decision Trees?
# Revisiting their Objectives, Tuning, and Performance

## Abstract

Recently there has been a surge of interest in optimal decision tree (ODT) methods that globally optimize accuracy directly, in contrast to traditional approaches that locally optimize an impurity or information metric. However, the literature shows conflicting evidence on the value of ODTs, with some demonstrating superior out-of-sample performance of ODTs over greedy approaches, while others show the opposite. The value and performance of ODTs therefore remains one of several open question regarding ODTs, most of which could not be answered before due to lack of scalability. With our experimental study—the largest to this date—we examine five such open questions. Our results show (i) that a major advantage of ODTs over greedy approaches is that they can optimize the target objective directly (e.g., accuracy rather than a proxy such as Gini impurity); (ii) that hyperparameter tuning of ODTs is essential; and reaffirm (iii) that optimal methods, on average, obtain smaller and more accurate trees than greedy approaches. Our results also refute two previously posited hypotheses: (iv) that the difference between optimal and greedy approaches diminish with more data, and (v) that optimal methods are more sensitive to overfitting. Finally, our work provides insights on the value of ODTs, clear recommendations for researchers and practitioners on the usage of greedy and optimal methods, and code for future comparisons.

## 1 Introduction

Decision trees (DTs) are among the most-used (interpretable) machine learning models. Despite their simplicity, they can learn complex non-linear relationships in data and their human comprehensibility answers the need for interpretable models in high-stake domains (Rudin, 2019; Arrieta et al., 2020), provided the trees are small. *Optimal decision trees* (ODTs) specifically, which provably optimize an objective for a given size limit, provide small but accurate models (in particular for tabular data) and thus combine high performance with interpretability (Loh, 2014; Piltaver et al., 2016; Carrizosa et al., 2021).

However, since training optimal decision trees with respect to a size limit is NP-hard (Hyafil & Rivest, 1976), most early decision tree learning methods were greedy top-down induction heuristics. Such methods, like CART (Breiman et al., 1984) and C4.5 (Quinlan, 1993), locally optimize some impurity or information gain metric for each branching node in linear time, which makes learning a tree as fast as sorting the data. Despite their simplicity, these greedy models perform remarkably well and are still the backbone of many state-of-the-art learning methods, such as boosting and random forests (Grinsztajn et al., 2022). Consequently, *greedy* decision tree learning has been extensively studied.

In contrast, *optimal* decision tree research is a much younger and less understood field. Lack of scalability prevented early adoption, and also prevented an extensive investigation of the differences between optimal and greedy learning. For example, while Murthy & Salzberg (1995) investigated the effectiveness of the greedy approach, they lacked a scalable ODT method to compare to and therefore confined their analysis on small synthetic data. More recently, Bertsimas & Dunn (2017) proposed to use mixed-integer programming (MIP) to learn ODTs, but again, lack of scalability confined their analysis to datasets of only 100 instances or trees with a maximum depth of two, while for larger problems, their approach did not converge to optimality. Therefore, the support for several of their claims remains uncertain.

To remedy this, recent research focused on improving the scalability of learning ODTs, to reduce runtimes and support learning on larger datasets by using techniques such as Boolean satisfiability (Shati et al., 2023) and dynamic programming (e.g., Demirović et al., 2022; Brita et al., 2025), resulting in runtime improvements up to several orders of magnitude. Due to these algorithmic advancements and an increase in computation power, we can now use a recent dynamic programming approach (Van der Linden et al., 2023) to analyze datasets with up to hundreds of thousands of instances.

Therefore, in this paper, we use this new-found scalability to study the following five open questions about optimal decision trees that as of yet are either left unexplored or where previous investigations provided unclear, indecisive, and even conflicting answers.

**Open Question 1**: *What is the effect of training the traditional decision tree objectives such as Gini impurity and information gain to optimality?*

Traditionally, decision trees are optimized using an information (entropy) (Quinlan, 1986) or Gini impurity (Breiman et al., 1984) objective, motivated by information theory and probabilities of misclassification respectively. Others have suggested optimizing the minimum description length (Quinlan & Rivest, 1989; Mehta et al., 1995) or finding a maximum a posteriori tree (Denison et al., 1998) and many other objectives. Optimal decision trees, on the other hand, typically optimize accuracy directly. However, some believe that we should not directly optimize accuracy, and argue instead for optimizing the objectives that are commonly used for greedy learning, such as Gini impurity (Liu, 2022) and the Bayesian posterior (Sullivan et al., 2024).

Therefore, we investigate the effect of six commonly used greedy learning objectives on ODTs and compare them with optimizing for accuracy. Our analysis shows that for ODTs learning the target objective directly is best. Greedy approaches, on the other hand, are confined to strictly concave objectives such as entropy (Kearns & Mansour, 1996). This therefore, reveals one of ODTs' benefits over greedy learning, since it can optimize the target objective directly without having to define a proxy splitting criterion.

**Open Question 2**: *What is the effect of the hyperparameter tuning method on optimal decision trees?*

Since optimal decision trees maximize performance under a size limit or penalty, setting the appropriate limit through hyperparameter tuning is important. While complexity-cost tuning is the standard method for greedy learning, the literature has no standard approach for ODTs, nor a comparison of the existing approaches. For example, Nijssen & Fromont (2007) tune a minimum support constraint, Aglin et al. (2020) tune only the tree depth, Lin et al. (2020) tune the complexity cost, Demirović et al. (2022) tune both the depth and the number of nodes, and some use no hyperparameter tuning at all (e.g., Marton et al., 2024).

Therefore, we investigate the effect of hyperparameter tuning for ODTs. Our experiments show that for optimizing accuracy, hyperparameter tuning is essential, but in most cases, the specific approach is not. The choice of hyperparameter tuning can have an impact on the size of the trees and on the total runtime.

**Open Question 3**: *Are optimal decision trees more accurate than greedy decision trees?*

One of the main claims by Bertsimas & Dunn (2017) is that ODTs "give average absolute improvements in out-of-sample accuracy over CART of 1-2%." Similar results were also reported by, e.g., Lin et al. (2020); Demirović et al. (2022) and Mazumder et al. (2022). However, this claim is contested by, for example, Zharmagambetov et al. (2021) and Sullivan et al. (2024), who observe *worse* results for ODTs compared to CART, and, as a consequence, doubt whether these trees can really be called "optimal".

We analyze the experimental setups of all previous comparisons between ODTs and CART (that we are aware of) and conclude that the main difference that explains the conflict is whether all learning methods were run with or without a depth limit. The claim by Bertsimas & Dunn (2017) should therefore be corrected by including the condition "for a given depth limit." Even then, because the effect of the depth limit may differ across datasets, the original claim may not be true. Therefore, we instead investigate the alternative claim that *ODTs have a better accuracy-interpretability trade-off* (Lin et al., 2020), where we use tree size (number of leaves) as a proxy for interpretability. Our empirical analysis shows this claim to be true. Thus, the core value of ODTs is their compactness (i.e., their interpretability), while being competitive in accuracy.

**Open Question 4**: *Do the differences between greedy and optimal decision trees diminish with more data?*

In one of the most important early comparisons between greedy and optimal decision trees, Murthy & Salzberg (1995) observe that the greedy heuristic gets increasingly more accurate with more (noise-free) training data, and therefore resembles the ground truth (optimal) decision tree more closely. From this observation, Costa & Pedreira (2023) raise the hypothesis that the gap between greedy and optimal decision trees diminishes with more data. As of yet, the validity of this hypothesis is an open question.

However, the observation by Murthy & Salzberg (1995) only referred to the accuracy of optimal and greedy trees, and they also only considered greedy trees without a size limit. Therefore, to further investigate the differences, we compare ODTs with greedy decision trees trained both with and without a depth limit on datasets with increasingly more training instances, and measure both accuracy and tree size. Our results show that in both cases the differences between greedy and optimal trees actually increase rather than decrease. For a fixed depth limit, the greedy tree fails to recover the optimal tree even as data increases. Without a depth limit, the accuracy of the greedy tree approaches or exceeds the optimal tree, but the tree size never approaches the optimal tree size.

**Open Question 5**: *Are optimal decision trees more prone to overfit than greedy decision trees?*

Dietterich (1995) argued against using optimal decision trees because they are prone to overfit. Recent work shows a similar concern, such as Blanc et al. (2023) who seek to find the trade-off between greedy and optimal learning and hypothesize that optimal learning tends to suffer from overfitting. Similarly, Sullivan et al. (2024) also claim that ODTs often suffer from overfitting.

However, these observations are based on comparing ODTs without proper hyperparameter tuning of their size with greedy approaches for which the size is established through proper hyperparameter tuning. When we investigate this claim with proper hyperparameter tuning of ODTs on both real and synthetic data with noise, we conclude that ODTs are actually less prone to overfit than their greedy counterparts.

**Contributions.** Our contribution is the largest evaluation to this date on optimal and greedy decision tree methods, on 109 real-world and additional synthetic datasets (small and large), and trees that go beyond small depth limits. We use this analysis to answer the five open questions above, and also formulate best practices for future comparisons.[1]

The remainder of the paper is organized as follows. Section 2 provides a general introduction to greedy and optimal decision tree learning. Section 3 reviews previous comparisons between the two. In Section 4, we evaluate the five open questions listed above. Section 5 provides practical recommendations for the use of and comparison with optimal decision tree methods and Section 6 draws an overarching conclusion. To keep the scope of this study manageable, we chose to limit this study to binary classification trees with hard axis-aligned splits, which are arguably the most common type of decision trees.

## 2 Background on Greedy and Optimal Decision Tree Learning

Decision trees are simple, interpretable machine learning models that can be used to predict non-linear relations. For example, the decision tree in Fig. 1a accurately predicts the data in Fig. 1b and can be easily interpreted by following the paths from the root node to each leaf. A major challenge in the field is devising algorithms that efficiently and effectively learn decision trees from data. This paper compares two commonly used approaches: greedy top-down induction and optimal approaches. Historically, the challenge for greedy approaches has been learning small trees, whereas for optimal approaches the challenge was scalability.

**Greedy top-down induction.** Decision tree learning started decades ago with AID (Morgan & Sonquist, 1963), a recursive approach to regression analysis, later adapted for classification in CHAID (Kass, 1980). Since then, two of the most popular decision tree learning algorithms have been CART (Breiman et al., 1984) and ID3 (Quinlan, 1986), with its successor C4.5 (Quinlan, 1993). Each of these uses top-down induction (TDI) to greedily partition the data by finding a split that is locally optimal according to an information or impurity criterion. High-level code for a greedy decision tree learner is given in Algorithm 1.

---

[1]All code is provided to the reviewers and will be made public on acceptance.

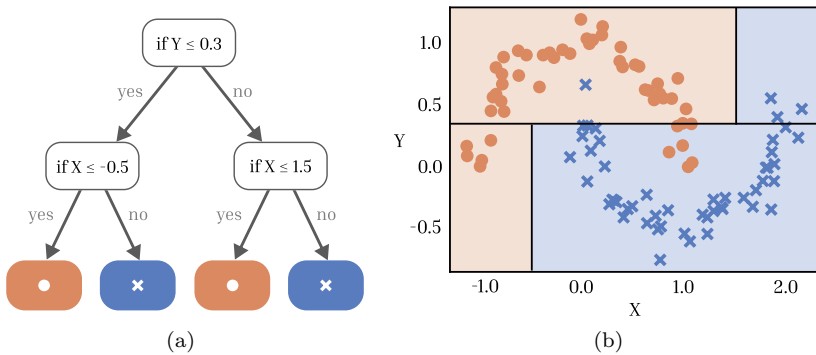

Figure 1: A decision tree with four leaves and its predictions on the moons dataset. Size-limited decision trees are easily interpretable by following the prediction paths from the root to each leaf.

---

**Algorithm 1:** High-level code for learning greedy decision trees based on recursion.

---

Given instances $X$ with target classes $Y$, tree $= greedy(X, Y)$
**function** *greedy(X, Y)* **is**
    **if** *depth/size limit reached or all instances $Y$ are the same class* **then**
        **return** Leaf predicting the majority class in $Y$
    Find the best feature $\phi$ and threshold $\tau$ to split on (minimizing splitting criterion)
    Partition $(X, Y)$ into $(X_L, Y_L)$ and $(X_R, Y_R)$ by testing for each instance $(x, y)$ in $(X, Y)$ if $x_\phi \leq \tau$
    **return** Node with predicate $x_\phi \leq \tau$, left child $greedy(X_L, Y_L)$, right child $greedy(X_R, Y_R)$

---

A benefit of greedy decision tree learners is that they can be efficiently implemented by sorting feature values in $O(n \log n)$ time (for $n$ instances) and then scanning them in linear time for each layer of the tree. Unfortunately, while this greedy procedure is fast, it can, in theory, produce decision trees that are arbitrarily larger than optimal (Garey & Graham, 1974).

**Optimal approaches.** In contrast to greedy learning, optimal decision tree (ODT) learners find a decision tree that globally maximizes some prediction objective on the training set. For example, many ODT classification algorithms directly maximize predictive accuracy combined with a hard or soft constraint on the tree size. Many ODT approaches also make it relatively easy to optimize other objectives or add constraints (Nijssen & Fromont, 2010; Verwer & Zhang, 2017). However, ODTs are much more expensive to compute since the problem of identifying optimal decision trees is NP-hard (Hyafil & Rivest, 1976; Ordyniak & Szeider, 2021).

Over the past few decades, many algorithms for finding ODTs with increasingly better scalability have been proposed, based on techniques such as mixed-integer programming (e.g., Bertsimas & Dunn, 2017; Verwer & Zhang, 2017; Aghaei et al., 2024), constraint programming (Verhaeghe et al., 2020), SAT solvers (e.g., Narodytska et al., 2018; Hu et al., 2020; Shati et al., 2023), continuous optimization (e.g., Blanquero et al., 2021), and dynamic programming (e.g., Aglin et al., 2020; Lin et al., 2020; Brita et al., 2025). While the first algorithms for ODTs could only find optimal models for datasets with a few hundred instances, state-of-the-art dynamic programming algorithms enable us to train ODTs with 100,000s of instances. In this work, we therefore base our empirical experiments on the dynamic-programming-based STreeD algorithm (Van der Linden et al., 2023).

The dynamic programming (DP) approach obtains better scalability by exploiting two key observations: (i) optimal solutions to subtrees are independent of one another, and (ii) solutions to repeated subproblems, defined by the dataset $X, Y$ and the size budget (e.g., the depth budget $d$), can be cached and reused (Nijssen & Fromont, 2007). Further scalability improvements to the dynamic programming approach include branch-and-bound pruning (Aglin et al., 2020), similarity-based pruning (Lin et al., 2020; Brita et al., 2025), and

faster computation of small optimal subtrees (Demirović et al., 2022; Brita et al., 2025). The core optimal recursive formulation (without the improvements just listed), is captured in Algorithm 2.

---

**Algorithm 2:** High-level code for optimal decision trees based on recursion.

---

Given instances $X$ with target classes $Y$, and depth budget $d$, tree $= opt(X, Y, d)$
**function** *opt(X, Y, d)* **is**
    **if** $d = 0$ *or all instances $Y$ are the same class* **then**
        | **return** Leaf predicting the majority class in $Y$
    **if** $(X, Y, d)$ *in* cache **then return** cache$(X, Y, d)$
    $T^* \leftarrow$ Leaf predicting the majority class in $Y$
    **for** *each feature $\phi$ and threshold $\tau$ to split on* **do**
        Partition $(X, Y)$ into $(X_L, Y_L)$ and $(X_R, Y_R)$ by testing for each instance $(x, y)$ in $(X, Y)$ if $x_\phi \leq \tau$
        $T \leftarrow$ Node with predicate $x_\phi \leq \tau$, left child $opt(X_L, Y_L, d-1)$, right child $opt(X_R, Y_R, d-1)$
        **if** $T$ *has a lower error than $T^*$ on instances $X, Y$* **then** $T^* \leftarrow T$
    cache$(X, Y, d) \leftarrow T^*$
    **return** $T^*$

---

**Decision tree variants.** In this work, we aim to study decision trees that split on one feature at a time (axis-aligned) and create binary splits based on a 'less-than-or-equal' relation, since these are indisputably the most commonly used decision trees. However, it is worth noting that there are other variants of decision trees. For example, an oblique decision tree splits on a linear combination of feature values inside each node (e.g., Boutilier et al., 2023); multi-split trees branch a node into more than two subtrees (e.g., Chaouki et al., 2024); and randomized decision trees consider 'soft' splits such that instances close to the cut appear in both subtrees with a weight based on its distance from the cut (e.g., Blanquero et al., 2021). Also, we limit this paper to classification trees, which are arguably the most studied decision trees, and refer to Van den Bos et al. (2024) for a recent study on optimal regression trees.

## 3 Previous Comparisons between Greedy and Optimal Methods

This section reviews previous comparisons of optimal and greedy decision tree learning methods, each different in its experiment design and sometimes in its conclusion. We highlight three of them—those by Murthy & Salzberg (1995), Bertsimas & Dunn (2017), and Zharmagambetov et al. (2021)—while we review all other previous comparisons (that we know of) in Appendix A, for which we provide only a summary here. Finally, we discuss good and poor comparison practices as input for a fair comparison method presented after.

**Highlights of three previous comparisons.** Murthy & Salzberg (1995) compare the greedy approach to known true synthetically generated optimal trees. They observe that the greedy tree is approximately one standard deviation larger than the true tree size while the expected depth (the average path length over all instances) is very close to optimal, which was also observed by Goodman & Smyth (1988). However, the maximum depth of greedy trees is on average approximately two times higher than the true depth. When they increase the dataset size linearly with the true tree size, they observe almost no drop in accuracy for the greedy approach. They conclude that "the more (noise-free) training data there is, the more accurately and reliably greedy induction can learn the underlying concept" (i.e., resemble the ground truth tree). From this result, Costa & Pedreira (2023) hypothesize that the gap between optimal and greedy approaches diminishes for more data (Open Question 4). However, Murthy & Salzberg (1995) had no access to a scalable ODT method and therefore only compared greedy trees with synthetically generated ground truth trees.

Two decades later, Bertsimas & Dunn (2017) compared their optimal method OCT with CART on both real and synthetic data. When CART is constrained to the same depth limit as OCT (up to depth four), they conclude that OCT, on average, has a statistically significant 1-2% better out-of-sample accuracy (Open Question 3). The largest difference is observed at depth two. They hypothesize that the smaller difference

| Year | Author(s) | Method | Greedy with same size limit | Greedy without size limit | Small and large datasets | Beyond small trees | Optimal tuned (correctly) | Greedy tuned (correctly) |
|---|---|---|---|---|---|---|---|---|
| *Papers that propose ODT methods* | | | | | | | | |
| 2007 | Nijssen & Fromont | DL8 | ✓ | ✓ | | ✓ | | |
| 2017 | Bertsimas & Dunn | OCT | ✓ | ✓ | | ✓ | ✓ | ✓ |
| 2019 | Verwer & Zhang | BinOCT | ✓ | | | ✓ | | |
| 2020 | Lin et al. | GOSDT | | | | ✓ | ✓ | |
| | Hu et al. | MaxSAT | ✓ | | | ✓ | | |
| 2021 | Günlük et al. | ODT | ✓ | | | | ✓ | ✓ |
| 2022 | Demirović et al. | MurTree | ✓ | | ✓ | ✓ | | |
| | Hua et al. | RS-OCT | ✓ | | ✓ | ✓ | ✓ | ✓ |
| | Mazumder et al. | Quant-BnB | ✓ | | ✓ | | | |
| 2024 | Liu et al. | BNP-OCT | ✓ | | ✓ | | | |
| | Alès et al. | CTT | | ✓ | | ✓ | ✓ | ✓ |
| 2025 | Brita et al. | ConTree | ✓ | | ✓ | ✓ | ✓ | ✓ |
| *Other papers that compare ODTs with greedy DTs* | | | | | | | | |
| 1995 | Murthy & Salzberg | - | | | | ✓ | | ✓ |
| 2021 | Zharmagambetov et al. | - | | ✓ | ✓ | ✓ | | ✓ |
| 2024 | Sullivan et al. | MAPTree | | ✓ | | ✓ | | |
| | Marton et al. | GradTree | | ✓ | ✓ | ✓ | | |
| 2026 | This work | - | ✓ | ✓ | ✓ | ✓ | ✓ | ✓ |

Table 1: Simplified overview of the comparisons between greedy and optimal methods in the literature. Ideally, a comparison checks all columns. (i) Compare methods under the same size constraint; (ii) compare (greedy) methods without a size constraint; (iii) compare on small and large datasets ($> 10.000$ instances); (iv) compare optimal methods beyond depth three; (v) tune optimal methods (correctly); and (vi) tune greedy methods (correctly).

for depths three and four is the result of OCT not converging to optimality within their time limit. When CART is run with a depth limit of ten, it is negligibly better than OCT at depth four.

However, the main restriction of their analysis is the scalability of OCT. Because of this, they restrict synthetic data analysis to datasets with only 100 instances and two continuous features. They also experiment with datasets up to 1600 instances, but only on ground truth trees of depth two. When training OCT on the synthetic data, they set the maximum depth to the true depth, which prevents overfitting.

Zharmagambetov et al. (2021) compare greedy methods with their local search method TAO (Carreira-Perpinán & Tavallali, 2018) and the optimal methods OCT and GOSDT (Lin et al., 2020). They conclude that most methods perform similarly, except TAO, which outperforms the other methods. In many cases, they observe that CART outperforms OCT and GOSDT. For CART and TAO, they train greedy trees up to depth 30. For OCT, they use the results reported previously by Bertsimas & Dunn (2017), which go up to depth four. They train GOSDT with a high complexity cost, yielding trees that are on average no larger than 3.4 leaf nodes for any of the datasets. They also observe that for most datasets, GOSDT runs into the two-hour time-out. Based on these results, they cast doubt on the name "optimal" decision trees and on the practical usability of ODTs.

**Summary of other comparisons.** We observe that the main difference between Bertsimas & Dunn (2017) and Zharmagambetov et al. (2021) is *whether they compare ODTs to CART with or without the same*

*depth limit.* Also in the other comparisons reviewed in Appendix A, we observe that most papers that propose new ODT methods aim to show ODTs' superior performance under a fixed depth limit. On the other hand, papers that propose decision tree methods other than optimal approaches, typically aim to find the method that obtains the best out-of-sample accuracy without imposing depth constraints on the tree.

Furthermore, we make the following two observations from the comparisons reviewed in Appendix A. First, *both greedy and optimal methods are often not correctly tuned, or not tuned at all.* When comparing with CART, many papers show several modifications to how CART is trained and tuned. We assess the impact of each of these modifications on the accuracy in Appendix B and suggest best practices on how to train CART. Additionally, several papers evaluated ODTs without tuning the complexity, which significantly worsens the ODT performance (Open Question 2), and caused researchers to believe ODTs are prone to overfitting (Open Question 5). In contrast, some others evaluate ODTs with a substantial restriction on the tree size, resulting in shallow underfitting trees.

Second, *many comparisons are limited to small datasets and small trees* (e.g., less than 10000 instances or trees of maximum depth three or even two). This is typically because of scalability limitations. The improved scalability of optimal methods allows us to analyze larger datasets and trees.

Table 1 summarizes our observations about previous comparisons. We recommend that future comparisons between ODTs and greedy approaches should check all the columns in this table. We summarize our recommendations as follows:

---

**Recommendations 1** (Greedy-Optimal Comparisons)**.**

1. Compare both with and without constraining the sizes of the decision trees.
   *You can compare ODTs with other decision tree learning methods without a size limitation if accuracy is the only concern. Comparing with the same size limitation in addition fairly compares the ability to learn small (interpretable) and accurate models.*

2. Compare performance on both small and large datasets.
   *While experiments on small data are more efficient to run, their results often do not carry over to larger datasets.*

3. Evaluate both small and large trees.
   *Several previous comparisons only compared trees of depth two. This optimization problem is too simplistic, and the results do not always carry to a larger depth.*

4. Tune both greedy and optimal methods and ensure a fair comparison.
   *Comparisons between greedy and optimal trees at a fixed depth can be unfair since the methods respond differently to size constraints. The best practice, therefore, is to tune the hyperparameters of both methods.*

---

# 4   Evaluating Five Open Questions about Optimal and Greedy Decision Trees

Using the established best practices in Section 3, this section empirically investigates the five open questions mentioned in the introduction. We first describe the experiment setup, then evaluate the five questions, and end with a note on the scalability of ODTs.

### Experiment Setup

To evaluate the five open questions, we compare optimal and greedy methods on both synthetic and real datasets. We describe below how we train ODTs, how we obtained the data, and what statistical analysis we use. Runtime results are obtained with an Intel(R) Xeon(R) Gold 6448Y CPU using at most 4GB of RAM.

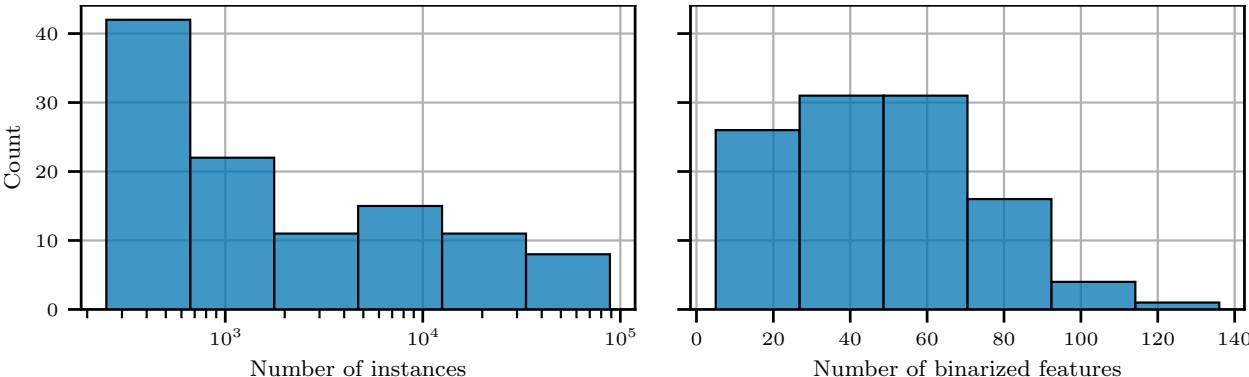

Figure 2: Histogram of the number of instances and features after binarization in the 109 OpenML datasets considered in our experiments. See Appendix G for the list of all datasets.

**Training methods and binarization.** We train ODTs while tuning the number of branching nodes using five-fold cross-validation.[2] We implemented all the ODT objectives in the ODT method STreeD (Van der Linden et al., 2023, note that we refer to this method simply as "ODT") because of its scalability and flexibility in supporting new objectives and tuning methods.[3]

As with most state-of-the-art ODT methods, STreeD depends on binarization of numeric and categorical features to obtain good scalability. Therefore, we binarize the numeric training data with thresholds on the ten quantiles, and the categorical data with one-hot encoding (with at most ten categories). The test data is binarized in the same way. We experimented with other binarization approaches, but noted no significant impact with regard to the analysis presented here. We evaluate both CART and ODT on the binarized data to eliminate this difference between the two methods and focus only on the difference between greedy and optimal search. In Appendix D, we use the ODT method ConTree (Brita et al., 2025) on a subset of the datasets to show that similar results hold if we compare both methods on the original numeric data without binarization. We train CART as described by the best practices in Appendix B. We use the scikit-learn implementation of CART,[4] except for Open Question 1, where we use our own implementation to support a variety of objectives.

**Real data.** For the real datasets, we obtained a large benchmark set from OpenML (Vanschoren et al., 2013; Feurer et al., 2021). For the sake of scalability, we selected all binary classification datasets with 50 or fewer features, of which eight or fewer numeric features and no large text features, with no missing values, with at most 100,000 instances, and at least 250 instances. We take the most recent version of the dataset and omit duplicates or datasets that only differ in the random seed, resulting in 109 datasets. We split each dataset into five folds, creating five train and test pairs each consisting of four and one fold respectively. We list all datasets used in Appendix G. Fig. 2 shows a histogram of the number of instances and the number of binarized features of the datasets considered in this paper.

**Synthetic data.** Additionally, we evaluate on synthetic datasets where we can control the amount of noise. We follow the synthetic data setup by Murthy & Salzberg (1995), Bertsimas & Dunn (2017) and Dunn (2018). We generate $n$ random training instances with $p$ numeric features, uniformly distributed over $[0, 1]$. For a given noise strength $f \in [0, 1]$, we add feature noise by adding noise uniformly drawn from $[-f, f]^p$. For the synthetic data, we binarize the numeric features by threshold predicates on 10 quantiles per numeric feature. We generate a random binary tree on this binarized data of a maximum depth $d$ with at most $2^d$ leaf nodes. We choose random splits on the data such that each leaf node contains at least five instances. The binary labels of each leaf node are assigned alternately, such that no split leads to two leaf nodes with the same label. After this, we add class noise to a given percentage $c$ of the data by flipping its

---

[2]We consider a leaf node a depth-zero tree, so a depth-four tree has at most 16 leaf nodes and 15 branching nodes.
[3]https://github.com/algtudelft/pystreed
[4]https://scikit-learn.org/stable/modules/generated/sklearn.tree.DecisionTreeClassifier.html

label. For each training set, we create a corresponding test set without noise of 1000 instances per leaf node in the generated tree.

Unless otherwise specified, we set the true tree depth to three, the number of instances $n = 1000$, the number of numeric features $p = 3$, the feature noise $f = 0$, and the class noise $c = 0\%$. We test with changing the number of instances ($n = 50, 100, 250, 1000, 10000$), the number of features ($p = 2, 4, 6, 8$), the amount of feature noise ($f = 0, 0.2, 0.4, 0.6, 0.8, 1.0$), and the amount of class noise ($c = 0\%, 10\%, 20\%, 30\%, 40\%, 50\%$). We repeat each configuration 1000 times and report averages over these 1000 runs.

Additionally, we add synthetic datasets with a linear separator instead of a tree as the ground truth. The weights of the linear separator are chosen randomly from a normal distribution. Here too, we repeat each configuration 1000 times and report averages over these 1000 runs.

In our synthetic tree experiments, apart from the familiar test accuracy and number of leaf nodes, we also measure the following:

*True Discovery Rate (TDR):* The TDR is the percentage of splits in the ground truth tree that are recovered in the trained tree (higher is better).

*False Discovery Rate (FDR):* The FDR is the percentage of the splits in the trained tree that are not part of the ground truth tree (lower is better).

**Rank-based comparison.** Since accuracy scores across datasets cannot directly be summed and compared, we use the average accuracy rank as our main performance metric: for each dataset split, we rank all methods based on their respective test accuracy. If multiple methods have the same accuracy, they are all assigned the average rank. E.g., if two methods have the same best score, they both get rank 1.5. We then report the mean rank over all datasets. Lower mean rank is better.

Based on the best practices described by Demšar (2006), we compare the average rank of each method using a Nemenyi critical distance rank test. This test computes the critical distance (CD) between the average ranks of two methods to be statistically significant for a given confidence level (which we fix to $\alpha = 0.05$). For this test, we first compute the average rank per dataset, and then perform the test, since we cannot assume that the scores per dataset fold are independent.

### Open Question 1. Impact of the Objective

*What is the effect of training the traditional decision tree objectives, such as Gini impurity and information gain, to optimality?*

Traditionally, greedy decision trees have been trained with top-down induction that optimizes a local splitting criterion. The most commonly used are Gini impurity and entropy (or information gain), while some have also proposed to use the minimum description length principle (Quinlan & Rivest, 1989) or the tree with maximum likelihood given some priors (Chipman et al., 1998). On the other hand, most ODT methods directly optimize accuracy. Some, however, argue that optimizing Gini impurity (Liu, 2022) or the Bayesian posterior (Sullivan et al., 2024) is better than optimizing accuracy directly. Therefore, we empirically test the effect of the objective for ODTs on the 109 real datasets.

**Objectives.** We compare the out-of-sample accuracy of optimal and greedy decision trees (max-depth = 4) optimized with commonly used objectives. We describe each objective below as a leaf node error function that for a given number of instances $n$ and a number of misclassifications $m$ returns the objective value $f(n, m)$. Since we describe the function at the level of the leaf, the scores are not normalized. In the objectives evaluated here, we also do not consider any cost related to the complexity of the tree. We evaluate the following objectives:

*Accuracy:* Maximizing the training accuracy is the same as minimizing the number of misclassifications in each leaf node, and therefore
$$f_{\text{Accuracy}}(n, m) = m\,.$$

*Gini impurity:* Weighted Gini impurity scores are obtained by multiplying the Gini impurity by the number of instances in that leaf node. Let $p_0 = \frac{m}{n}$ denote the probability of the first class and $p_1 = \frac{n-m}{n}$ the probability of the second class. Then the objective value is

$$f_{\text{Gini}}(n, m) = n(1 - p_0^2 - p_1^2).$$

*Entropy:* Weighted scores are again obtained by multiplying by the size of the leaf node:

$$f_{\text{Entropy}}(n, m) = -n(p_0 \log_2 p_0 + p_1 \log_2 p_1).$$

*Minimum error:* Niblett (1987) estimates the expected error for nodes by assuming that every class has equal probability. It depends on the number of labels $|\mathcal{K}|$, and the count of the majority label $n_c$. In binary classification $|\mathcal{K}| = 2$ and $n_c = n - m$. Therefore, the expected error is

$$f_{\text{MinError}}(n, m) = n \frac{n - n_c + |\mathcal{K}| - 1}{n + |\mathcal{K}|} = \frac{n(m+1)}{n+2}.$$

Note that this is equivalent to Laplace smoothing with a smoothing parameter set to one (add-one smoothing) (Flach, 2012).

*Binomial pessimistic error (Binom.):* The commonly used C4.5 method (Quinlan, 1993) uses a pessimistic error by considering a leaf with $n$ training instances and $m$ misclassifications as a 'sample' from a binomial distribution with an unknown misclassifying probability. Since this probability cannot be computed directly, the upper confidence bound of the posterior distribution of this probability, based on a confidence level $\alpha$, is used as the error rate of the leaf node. The confidence interval width $z_\alpha$ is the $z$-value from the normal distribution for confidence level $\alpha$. Let $m' = m + \frac{1}{2}$ be the pessimistic error. Then the binomial pessimistic error can be expressed as:

$$f_{\text{Binom}}(n, m) = \begin{cases} n \cdot \left(1 - e^{\ln(\alpha)/n}\right) & \text{if } m = 0 \\ m & \text{if } m = n \\ \frac{m' + \frac{z_\alpha^2}{2} + \sqrt{z_\alpha^2 \left(m'\left(1 - \frac{m'}{n}\right) + \frac{z_\alpha^2}{4}\right)}}{n + z_\alpha^2} \cdot n & \text{otherwise.} \end{cases}$$

In our experiments, we use the same default $\alpha = 0.25$ as C4.5.

*Minimum description length:* The minimum description length approach states that the best model can be described with the least amount of bits of information because the description length of a model can directly be linked to the posterior probability of a model (Rissanen, 1978; Li & Vitányi, 2008). In practice, the encoding typically consists of two parts: first the encoding of the model and then the encoding of the data that deviates from the model. Here, we investigate the encoding proposed by Quinlan & Rivest (1989) who observe that the cost $L$ of encoding a binary string of length $n$ with $m$ ones and $n - m$ zeros can be computed by first encoding the size $n$ of the string, and then the positions of the $m$ ones, with $b$ representing an upper bound on the number of ones that can occur, i.e., $L(n, m, b) = \ln(b + 1) + \ln\binom{n}{m}$. Then, for every leaf node with $n$ instances and $m$ misclassifications, encode a bit string that specifies the misclassifications with cost $L(n, m, \lfloor(n+1)/2\rfloor)$ for binary classification:

$$f_{\text{MDL}}(n, m) = L(n, m, \lfloor(n+1)/2\rfloor).$$

Since we only focus on the leaf node cost, we ignore the cost of encoding the branching feature and the leaf node label.

*Bayesian:* Decision trees are also commonly trained using a Bayesian approach (Chipman et al., 1998; Denison et al., 1998). These approaches find the maximum likelihood tree given some priors. We present the objective function used in the recent work by Sullivan et al. (2024). For the binary case, they assume that each leaf node can be represented by a Bernoulli distribution with parameter $\theta \in [0, 1]$. They assume $\theta \in \text{Beta}(\rho_0, \rho_1)$, the Beta distribution with parameters $\rho_0, \rho_1 \in \mathbb{R}^+$. The values $\rho_0$ and $\rho_1$ are hyperparameters, but they fix these values to $\rho_0 = \rho_1 = 2.5$. The leaf error can then be expressed in terms of the Beta function $B$ as follows:

$$f_{\text{Bayes}}(n, m) = \ln B(\rho_0, \rho_1) - \ln B(m + \rho_0, n - m + \rho_1).$$

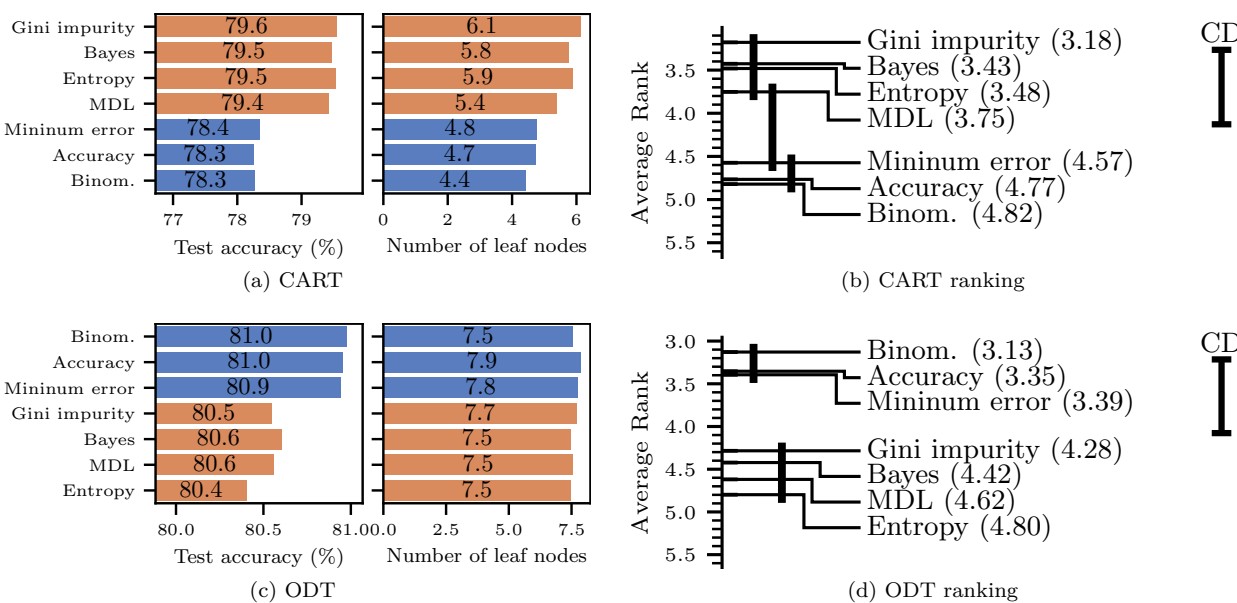

Figure 3: Comparing objectives for CART (a & b) and ODT (c & d) for max-depth = 4. For ODT we tuned the number of branching nodes and for CART we tuned the complexity cost. (a & c) Orange (blue) indicates (non-)concave. The average out-of-sample accuracy and the number of leaf nodes across all datasets and folds are shown, sorted by average rank. (b & d) Nemenyi critical distance rank test over the 109 datasets. The average rank per objective is plotted, and objectives with a rank difference smaller than the critical distance (CD) at p-value 0.05 are grouped by a black bar. For CART, the traditional concave objectives are significantly better than accuracy and similar objectives, whereas for ODT, they are significantly worse.

**Results and discussion.** Fig. 3a and 3c show the out-of-sample accuracy and number of leaf nodes for greedy and optimal decision trees respectively, when trained with the objectives introduced above (and with tuning the size and complexity cost for ODT and CART respectively). Fig. 3b and 3d show the average rank across all 109 datasets (lower is better). We use a Nemenyi critical distance rank test to test the statistical significance of the differences. Methods that are grouped by a black bar do not show a statistically significant ($\alpha = 0.05$) difference, whereas any pair of methods that is not covered by a bar have a statistically significant difference in accuracy performance.

The results in Fig. 3a align with the known fact that top-down induction heuristics require strictly concave objectives (Kearns & Mansour, 1996) (see Appendix C for an intuition). To make this obvious, we have colored strictly concave objectives orange while the rest is colored blue. Within the class of strictly concave and not-strictly concave objectives, the results for both greedy and optimal decision trees are not significant. On the other hand, the difference between the two groups for greedy and optimal decision trees is clear: for greedy optimization, concave objectives, such as the traditional Gini impurity and entropy, are all better than the non-concave objectives, whereas for ODTs, the precise opposite is observed. The statistical test shows that both these differences are statistically significant.

The new insight from these results is that global optimization of the traditional concave objectives results in significantly worse out-of-sample accuracy than when train accuracy is optimized directly. This shows that the strict concavity of objectives, such as Gini impurity and entropy, is not an inherently necessary or desirable property, but a *limitation* imposed by the greedy top-down induction approach.

This also reveals a more general strength of optimal decision trees: *optimal decision trees are flexible in the choice of an objective.* For greedy top-down induction approaches, each new decision tree optimization problem requires the construction of a new splitting criterion as an ad-hoc proxy for the true objective. For classification, Gini impurity and entropy have been shown to work well in practice. However, for other objectives, the search of a good proxy has been less successful. For example, for cost-sensitive classification

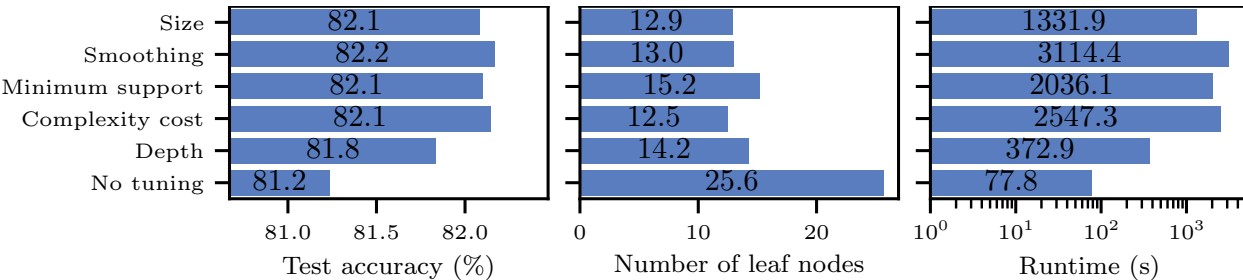

Figure 4: Complexity tuning results for ODTs with max-depth = 5 and for five runs on all 109 OpenML datasets. All tuning approaches result in approximately the same test accuracy.

Lomax & Vadera (2013) enumerate a large variety of cost-informed splitting criteria, but none of them are trivially obtained from the original objective nor is one splitting criterion clearly better than another. Similarly, to optimize a group fairness constraint, Kamiran et al. (2010) explore a variety of splitting criteria but notice no significant improvement with any of the suggested splitting criteria. On the other hand, both problems can be solved with ODTs by directly optimizing the target objective (Van der Linden et al., 2022; 2023), without having to find a good splitting criterion proxy first.

### Open Question 2. Impact of the Hyperparameter Tuning Method

*What is the effect of the hyperparameter tuning method on optimal decision trees?*

Recent work has proposed a variety of hyperparameter tuning methods for ODTs, but a comparison of all these approaches is as of yet lacking. For example, Nijssen & Fromont (2007; 2010) tune the *minimum support* constraint (the minimum number of instances required for each leaf), Aglin et al. (2020) and Mazumder et al. (2022) tune the *depth*, Lin et al. (2020) tune the *complexity cost* (the cost of adding a node), and Demirović et al. (2022) tune the *depth* and the *number of nodes*. In several papers, ODTs are also trained *without hyperparameter tuning* (e.g., Marton et al., 2024). As far as we know, no previous comparison measured the impact and the difference between these approaches, thus leaving the impact of hyperparameter tuning on ODTs as an open question.

**Tuning methods.** To test the impact of the tuning method, we compare them empirically by training optimal decision trees with max-depth = 5, while using five-fold cross validation to tune the hyperparameters defined by the method. For the sake of fairness, the number of configurations $k$ that can be tested with each method is fixed to ten.[5] We provide more details on how the values are chosen and experiments for other values of $k$ in Appendix F. All tuning methods are implemented in STreeD (Van der Linden et al., 2023). We evaluate the following five tuning methods:

*Depth:* We tune the depth parameter $d \in \{0, ..., \text{max-depth}\}$.

*Size:* We tune the number of branching nodes $n \in \{0, ..., 2^{\text{max-depth}} - 1\}$. The maximum value is always included and the other $k - 1$ options are selected using equal log spacing within this range (the values are evenly spaced after taking the logarithm).

*Complexity cost:* Complexity-cost tuning minimizes $\lambda |\mathcal{D}||L| + \sum_{(n,e) \in L} f(n, e)$, with $L$ the set of leaves, $|\mathcal{D}|$ the size of the dataset, and $\lambda$ the complexity-cost parameter. The minimum value for $\lambda$ that can result in a different tree is $\lambda = \frac{1}{|\mathcal{D}| \max\text{-depth}}$ because the worst-case scenario is that we need to add a whole new path of length max-depth to separate a single instance. We set the maximum value to 0.05 and select $k - 1$ options from this range using equal log spacing. The minimum step size between values is set to $\frac{1}{|\mathcal{D}| \max\text{-depth}}$. In all cases, we add $\lambda = 0$.[6]

---

[5] This is only the case for this experiment to make the comparison more fair. For the experiments for the other open questions, we tune the number of nodes of ODT without limiting the number of configurations, i.e., $k = 2^{\text{max-depth}}$.

[6] Lin et al. (2020) recommend for their method GOSDT to use $\lambda \geq \frac{1}{|\mathcal{D}|}$ for faster training, but this setting can exclude larger trees that are more accurate. Additionally, in their experiments, they aim to acquire trees of at most $n$ leaves by setting $\lambda = \frac{1}{n}$.

*Minimum support:* The *minimum support* is a hard constraint on the minimum number of instances required in a leaf node. We tune the minimum leaf node size $u$ as a percentage of the dataset size $|\mathcal{D}|$. The minimum value for $u$ is $\frac{1}{|\mathcal{D}|}$: precisely one instance should end up in each leaf node. For the maximum value, we compute the frequency of the majority class $|\mathcal{D}_{\text{majority}}|$ and set the maximum value of $u$ to $1 - \frac{|\mathcal{D}_{\text{majority}}|}{|\mathcal{D}|}$. Any value higher than this will always result in a single leaf node. We obtain $k$ values from this range using equal log spacing, with a minimum step size of $\frac{1}{|\mathcal{D}|}$.

*Smoothing:* The Laplace smoothing approach (Flach, 2012) assumes in a leaf node that for each class, $x$ extra instances exist. With $|\mathcal{K}|$ the number of classes, the accuracy objective becomes $f(n,m) = \frac{n(m+x)}{n+|\mathcal{K}|x}$. We select $k-1$ options for $x$ using equal log spacing from the interval $[\frac{1}{\text{max-depth}}, 0.05|\mathcal{D}|]$. Additionally, we always add the option $x = 0$. We also use $\frac{1}{\text{max-depth}}$ as the minimum step size.

**Results and discussion.** Fig. 4 shows the performance of the complexity tuning method on the OpenML datasets. Surprisingly, the results show that all tuning methods obtain similar accuracies: there is no statistically significant difference between any of the approaches. Therefore, we do not further test using a combination of the tuning methods. In terms of optimizing accuracy, the only conclusion is that using any of the tuning approaches is better than no tuning.

However, other differences can be observed between the methods. For example, tuning the minimum support or depth yields slightly larger trees because they do not directly constrain the number of nodes. Furthermore, the runtime results show that tuning only the depth of the tree is by far the fastest approach. We show and discuss further results in Appendix F.

We summarize the observations made from Open Questions 1 and 2 in the following recommendations.

---

**Recommendations 2** (How to train Optimal Decision Trees effectively)**.**

1. Optimize the same loss at train and test time for optimal decision trees.
   *Optimizing the accuracy on average yields the best out-of-sample accuracy. Avoid using concave proxies such as Gini impurity or entropy, since ODTs do not share the same limitations as greedy top-down induction approaches.*

2. Tune the complexity of optimal decision trees.
   *Training ODTs with hyperparameter tuning is significantly better than training without hyperparameter tuning.*

3. Tune the size, complexity cost, or smoothing parameter; or, in the case of large datasets, the depth.
   *Tuning the size, complexity cost, smoothing, or depth, on average, yields similar out-of-sample results. However, tuning the depth yields larger trees. For large datasets, tuning is less important, and tuning the depth is more runtime efficient.*

*These best practices are supported by the ODT learning method we use in our experiments: STreeD (Van der Linden et al., 2023).*

---

### Open Question 3. Out-of-Sample Accuracy

*Are optimal decision trees more accurate than greedy decision trees?*

Based on their results, Bertsimas & Dunn (2017) claimed that ODTs on average give "absolute improvements in out-of-sample accuracy over CART of 1-2%." Similar results were consecutively reported by, for

---

However, this also filters out many trees that have much less than $n$ leaves, since a single leaf node already has an accuracy greater than or equal to 0.5 for binary classification, so that even a perfect tree with $n$ leaves given such $\lambda$ would have a worse score than a single leaf node. Therefore, both of these settings are too conservative.

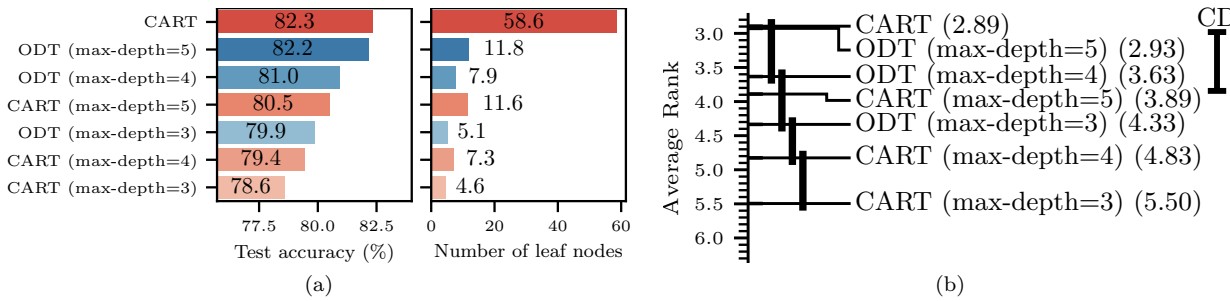

Figure 5: Out-of-sample accuracy of CART and ODT compared on five runs for all 109 datasets. (a) ODT (blue) versus CART (red). CART without a depth limit can obtain the same performance as ODT (max-depth = 5), but yields much larger trees. (b) Nemenyi critical distance rank test for ODT versus CART for the 109 datasets. The average rank per method is plotted and methods with a rank difference smaller than the critical distance (CD) at significance level $\alpha = 0.05$ are grouped by a black bar. With the same depth limit, ODT performs significantly better than CART.

example, Verwer & Zhang (2019), Lin et al. (2020), Mazumder et al. (2022), and Demirović et al. (2022). However, around the same time others drew the opposite conclusion from their experiments. For example, Zharmagambetov et al. (2021) and Sullivan et al. (2024) both report *worse* out-of-sample accuracy results for ODTs. As noted in Section 3, the difference in setup that explains the diverging results is whether (i) CART is trained with or without the same depth limit as ODT, and (ii) whether CART and ODT are trained according to the best practices. Therefore, we here compare the two approaches again, but now both with and without a depth limit for CART, and by adhering to the best practices for training both methods (see recommendations 1, 2, and 3). See Appendix D and E for the same experiment with numeric features and with balanced accuracy as the objective.

**Results and discussion.** Fig. 5 reports the average out-of-sample accuracy obtained by CART and ODT for a variety of depth limits over all 109 OpenML datasets. In these results, CART without a depth limit obtains the highest average test accuracy, albeit only marginally higher than ODT with a depth-five limit, and CART needs to train trees that are five times larger than ODT to obtain this result.

If, on the other hand, we compare CART and ODT with the same depth limit, Fig. 5 shows that for depth three, four, and five, *ODT obtains a significantly higher accuracy than CART*. For depth three, four, and five the average difference is 1.3%, 1.6%, and 1.7% respectively, aligned with the claim by Bertsimas & Dunn (2017). Since optimal algorithms optimize the complete decision tree, instead of greedily improving the tree, they can achieve better scores.

**Accuracy-interpretability trade-off.** The results in Fig. 5 reveal that the comparison of CART and ODT strongly depends on the selection of the depth limit, because with a sufficient depth limit, CART obtained the same test accuracy as ODT. However, the depth limit for which ODT obtains the same accuracy as CART without a depth limit may differ per dataset.

This reveals that the main advantage of ODT over CART is not better accuracy per se, but *better accuracy under the same size constraint*. Therefore, Lin et al. (2020) claim that the main advantage of ODTs is that they obtain a better accuracy-interpretability trade-off than CART (with tree size used as a proxy for interpretability). For example, consider the out-of-sample accuracy for ODT and CART for three datasets in Fig. 6 with increasing tree size limits. While ODT obtains higher accuracy for more restrictive size limits, the two methods converge to a saturation point where neither method can obtain a higher test accuracy with more nodes. That is, eventually both obtain the same accuracy, but ODT requires less nodes to do so.

A simple approach to measure both accuracy and size of a model is to compute the accuracy and subtract the size multiplied by some penalty term (see, e.g., Chaouki et al., 2024). The choice of this penalty term,

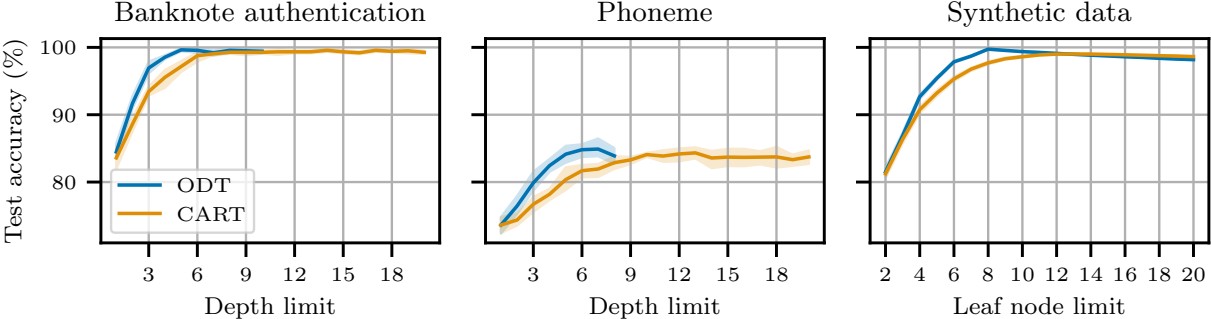

Figure 6: Typical accuracy-interpretability trade-off for untuned greedy and optimal decision trees. ODTs have an advantage for small size limits but both methods converge for large size limits.

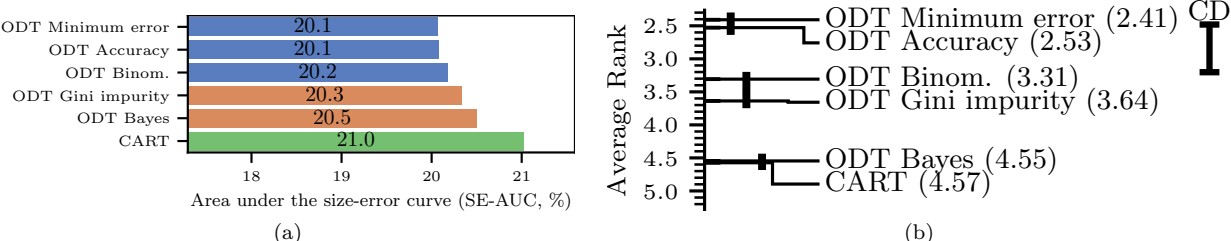

Figure 7: Comparing the test area under the size-error curve (SE-AUC) of several ODT objectives and CART on all 109 datasets. (a) The non-concave objectives are blue, the concave objectives are orange, and CART is green. (b) Nemenyi critical distance rank test. The average rank per method is plotted and methods with a rank difference smaller than the critical distance (CD) at significance level $\alpha = 0.05$ are grouped by a black bar. The ODT minimum error and accuracy are the best approaches. All ODT approaches except for Bayes are significantly better than CART.

however, is arbitrary. Moreover, this metric only provides a point estimate, whereas Fig. 6 shows that the performance difference changes for increasing size limits.

Therefore, we introduce a new metric called the *area under the size-error curve* (SE-AUC), which measures the error (100% minus the accuracy) for each size limit up to $s$ nodes and takes the average. More precisely, since we assume the end user will choose the size of the tree based on cross validation, we measure the Pareto-front of size and test accuracy: if a higher size limit yields a lower test accuracy, we continue measuring the area under the highest test accuracy seen so far. The choice of $s$ does not significantly impact the conclusion, since Fig. 6 shows that differences diminish for larger $s$. Therefore, increasing $s$ will change the absolute value of the SE-AUC metric but not the relative ranking. Since we use a rank-based test, the conclusion remains the same.

Formally, we define the SE-AUC$_s$ as follows:

**Definition 1** (Area under the size-error curve (SE-AUC)). Given a maximum number of leaf nodes $s$, the *area under the size-error curve* (SE-AUC$_s$) measures the area under the interpolated Pareto-front of size and accuracy up from 1 to $s$ nodes for a given method. It is measured by taking the following steps:

1. Learn a decision tree with $s$ different values for its complexity parameters (e.g., the complexity-cost parameter $\lambda$), so that, if possible, each value yields a differently sized tree. Record the resulting number of leaf nodes $i$ and test accuracy acc$_i$ for each run. If multiple runs yield the same number of leaf nodes, then average these test accuracies. (Note that not every learning method can directly set the output size of the tree.)

2. For missing tree sizes, linearly interpolate test accuracies using results from the nearest smaller and larger tree sizes. If the largest tree size obtained is less than $s$, extrapolate the test accuracy of this largest tree to all larger sizes up to $s$.

3. If larger trees result in smaller test accuracy, we replace it with the larger value for smaller trees, since we measure the Pareto front.

4. Finally, the average error (%) of the obtained trees of maximum size $s$ (number of leaf nodes): $\text{SE-AUC}_s = \frac{1}{s} \sum_{i=1}^{s} (1.0 - \text{acc}_i)$.

With this metric, we can now proceed to compare ODT with CART to test if ODT indeed obtains a better accuracy-interpretability trade-off than CART. Therefore, we train ODTs with the non-concave and the top two concave objectives from Open Question 1 and CART with the traditional Gini impurity objective on all 109 datasets. ODTs are trained with a maximum depth of four and CART is trained without a depth limit. For both methods, we train trees of one up to sixteen leaf nodes.

Fig. 7 shows that on average ODT achieves a significantly lower SE-AUC than CART, thus verifying that ODT has a better error-size curve than CART. With sufficient data, the training accuracy approaches the test accuracy for highly regularized models which means that optimal decision trees reliably improve over greedy trees. This result also repeats the conclusion of Open Question 1, that non-strictly-concave objectives are to be preferred over concave objectives for optimal methods.

**Open Question 4. Data Efficiency**

*Do the differences between greedy and optimal decision trees diminish with more data?*

Based on the results by Murthy & Salzberg (1995), Costa & Pedreira (2023) hypothesized that the differences between greedy and optimal decision tree learning diminish with more data. To test this hypothesis, we evaluate ODT and greedy performance on large real datasets and synthetic data with an increasing number of training instances and features.

**Result on synthetic data.** Fig. 8a shows how ODTs compare with CART for an increasing number of training instances on synthetic data generated from ground-truth trees of depth three. For less than 1000 instances, the ODTs are more accurate than both CART with and without a maximum depth limit. For more than 1000 instances, both ODT and CART obtain 100% test accuracy. CART, however, uses 11 leaf nodes to achieve this result, whereas the optimal approach only requires eight (equal to the true tree's complexity). Both approaches have approximately the same true discovery rate (TDR, percentage of ground truth splits that are found). However, ODT's false discovery rate (FDR, percentage of found splits that are not part of the ground truth) is lower. More instances help the ODT method to reduce its false discovery rate.

The depth-constrained CART's test accuracy plateaus around 1000 training instances at 98%. This shows that CART requires a higher depth limit to obtain the same accuracy as ODTs, regardless of how much data it receives. Even though the true tree depth is three, a maximum depth of four for CART is not enough to recover the tree.

Fig. 8b shows how CART's accuracy drops when the number of features in the synthetic data increases whereas the optimal approach retains the same accuracy. This difference can be explained by observing the rise in the FDR of CART when the number of features increases together with an increase in the number of leaf nodes: it finds more unnecessary splits. This shows that CART performs worse for an increasing number of features, whereas the optimal approach remains unaffected.

Fig. 8c additionally shows that when learning from synthetic data with a linear separator as the ground truth, CART without a depth constraint achieves higher accuracy than ODT, but with more data availability, CART also generates much larger trees with only a small gain in accuracy. Increasing the number of features in the synthetic data makes the classification function harder to learn. The relative accuracy performance of the methods stays roughly the same, but CART requires many more nodes. In all cases, ODTs perform better than CART with the same depth constraint.

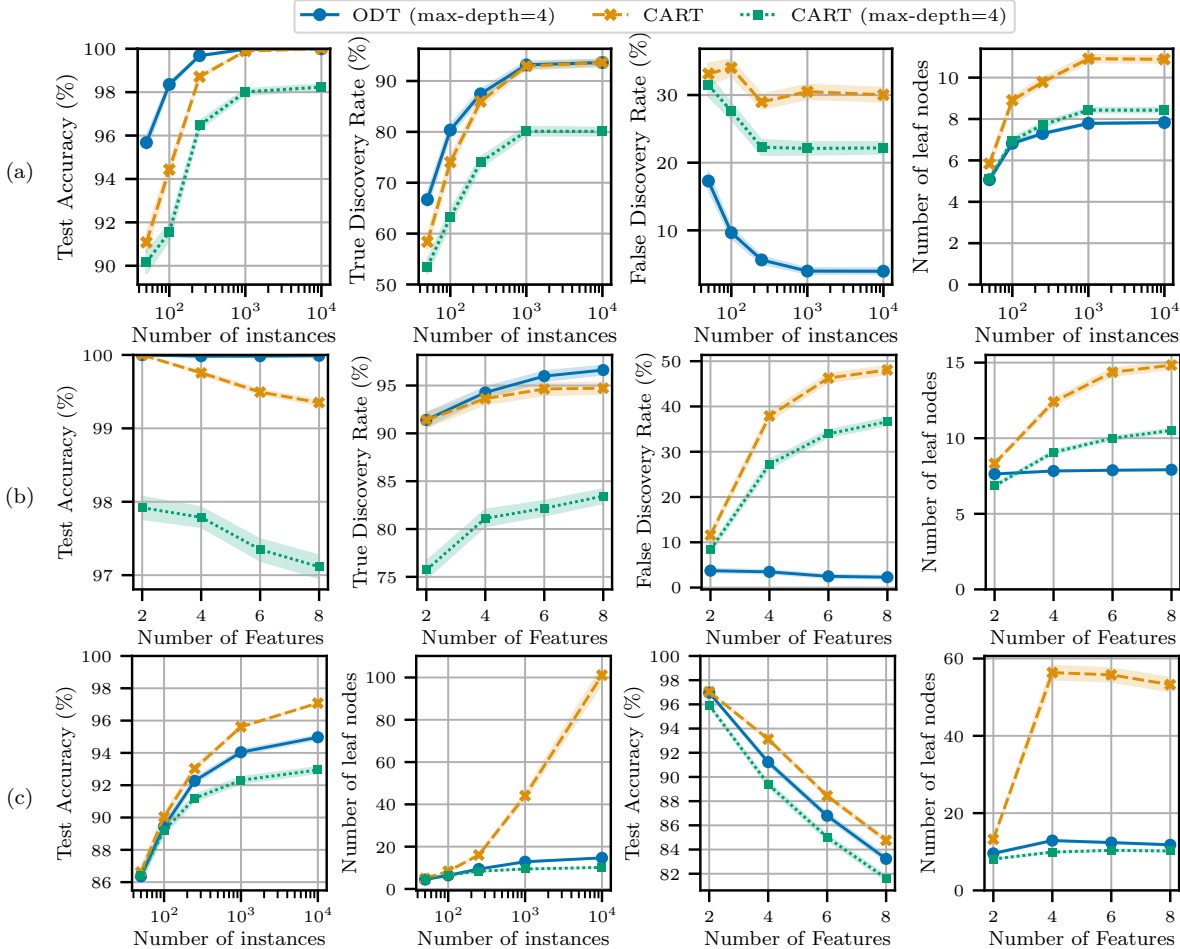

Figure 8: Results on the synthetic tree datasets for increasing (a) number of training instances, and (b) number of numeric features; and (c) on the synthetic linear datasets for increasing number of training instances and numeric features, to evaluate Open Question 4.

**Results on real data.** To evaluate Open Question 4 on real data, we consider the *Electricity* dataset from our benchmark, and two additional large datasets (*Covertype* and *Higgs*), chosen for their large number of instances (44,156, 566,602 and 940,160 respectively), to investigate performance under various dataset sizes.

Fig. 9 shows the out-of-sample accuracies for increasing number of training instances on these three large real datasets. We train ODTs with a depth limit of three, and CART with and without a depth limit of three. When the methods are *not* tuned, the optimal approach overfits on small datasets, obtaining a lower test accuracy than depth-limited CART. However, *with* tuning, this effect disappears and the ODTs' accuracy is consistently higher than CART's (depth limited). The difference in performance between tuning and not tuning diminishes for larger training sets. Without a depth limit, CART continues to increase its accuracy for more data. More data does not help depth-limited CART since, at some point, the greedy decisions on what feature and threshold to split on do not change anymore. In those cases, the added data does not lead to different greedy decisions but only makes them more certain.

However, Fig. 10 also shows that CART continues to grow larger trees with up to tens of thousands of nodes. For the Electricity dataset, for example, CART and ODTs have roughly similar accuracy for ten thousand training instances. However, the ODT has eight leaf nodes, whereas CART has over a hundred. These results contradict the observation by Oates & Jensen (1997) that greedy tree methods do not perform much better for more data but do yield larger trees with more data. For these datasets, we observe CART performing much better for more data while also resulting in larger trees.

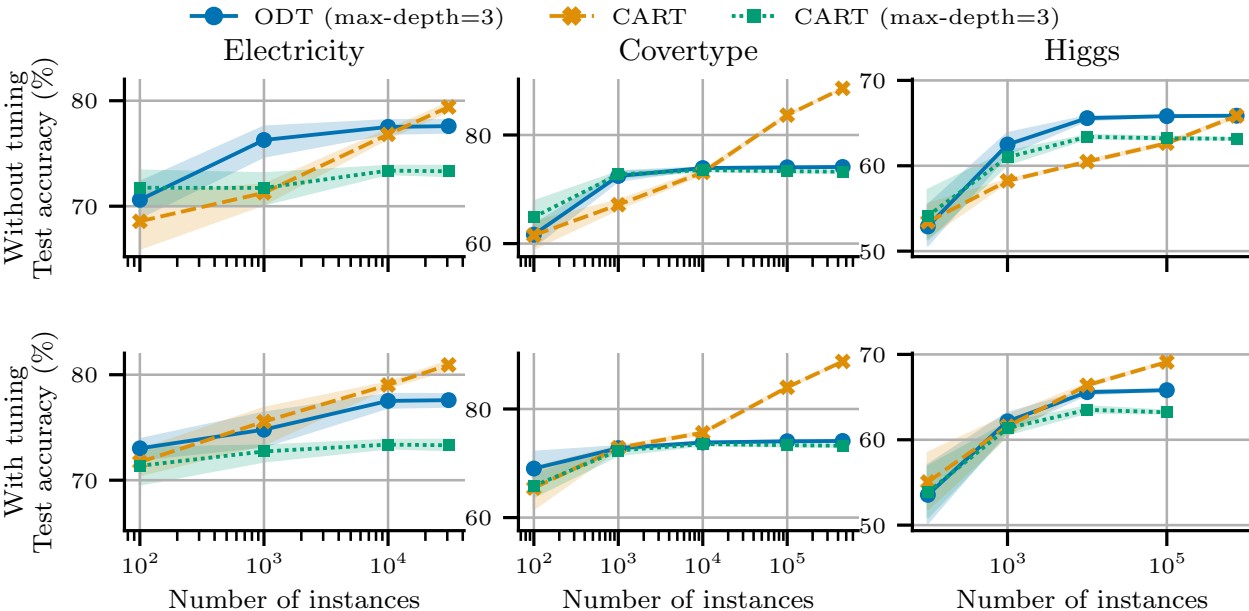

Figure 9: Test accuracies for increasing training instances with and without tuning.

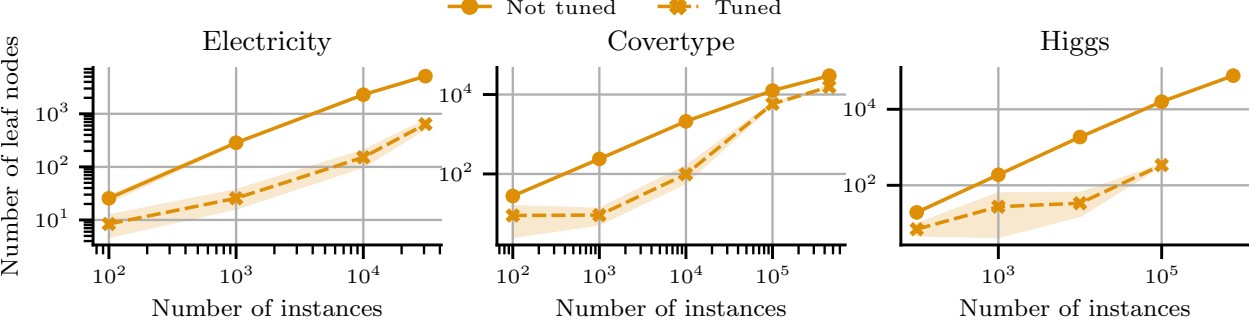

Figure 10: Number of leaf nodes for CART for increasing training instances with and without tuning. These numbers far exceed the eight leaf nodes for ODTs in Fig. 9.

**Discussion.** In conclusion, these results refute the hypothesis by Costa & Pedreira (2023) that the difference between optimal methods and greedy diminished for more data. CART (without a depth limit) can improve performance over ODT with sufficient data. However, unconstrained CART uses relatively uninformative splits and can result in trees that are orders of magnitude larger than ODTs. Depth-constrained CART may fail to recover an accurate tree even with large training sets and the difference with ODTs does not diminish. Therefore, for both depth-constrained and unconstrained CART, we find that they remain different from ODTs with more data.

## Open Question 5. Overfitting

*Are optimal decision trees more prone to overfit than greedy decision trees?*

A reoccurring critique on ODTs is that they are prone to overfit and more so than greedy trees. Initially, this concern was raised by Dietterich (1995), while more recently Blanc et al. (2023) and Sullivan et al. (2024) have raised similar concerns. On the other hand, as we have already seen in Open Question 3, with the same size limit, ODTs do obtain a significantly higher average out-of-sample accuracy, which argues against the concern for overfitting. Therefore, we analyze the risk of overfitting by comparing ODTs with greedy on

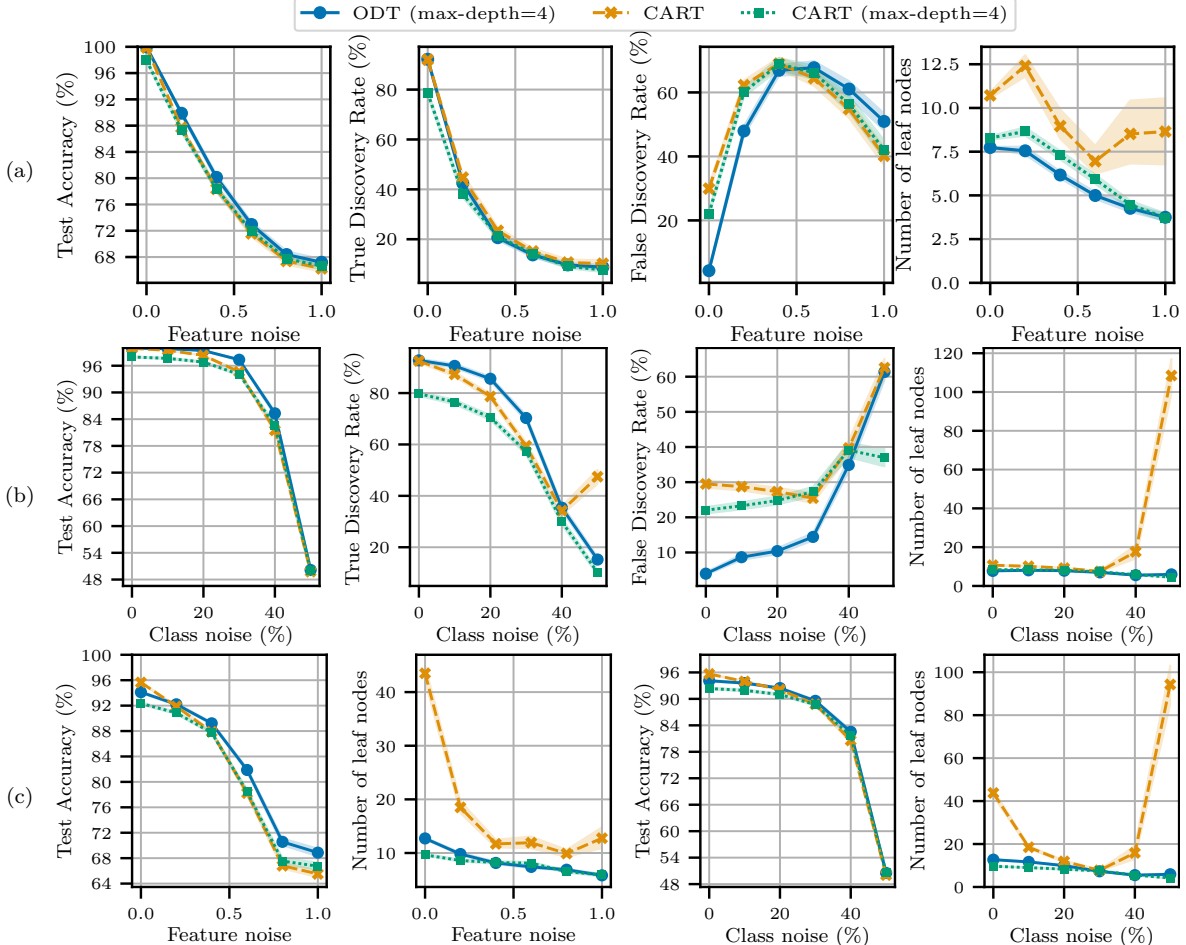

Figure 11: Testing Open Question 5 on the synthetic tree data for increasing (a) feature noise, and (b) class noise; and (c) on the synthetic linear data for increasing feature and class noise.

the synthetic data with noise, first by reiterating the earlier results on real data, and then by examining the performance of ODT and CART on synthetic data generated with noise.

**Overfitting on the real data.**   We already addressed overfitting on the real data above when observing the results in Fig. 9. These results showed that without hyperparameter tuning, ODTs are more prone to overfit than greedy approaches when data is sparse. However, with tuning and with the same size constraint, we observe that ODTs perform better than greedy trees on average.

**Overfitting on synthetic data with noise.**   Fig. 11a shows how both approaches respond to increasing feature noise in our synthetic data. For all amounts of feature noise, ODT obtains both a higher test accuracy and smaller trees than both CART approaches. The TDR and FDR are mostly similar, except for large amount of feature noise.

Fig. 11b shows that for increasing amounts of class noise, ODT's test accuracy is again consistently higher than both CART approaches. For large amounts of class noise, unconstrained CART obtains a lower test accuracy than both other approaches and also yields significantly larger trees. This result was obtained *with* five-fold cross-validation of the complexity cost of CART, as described in our best practices in Appendix B. To prevent generating such large CART trees, we have also experimented with fostering smaller trees by selecting the largest complexity cost that yields trees with a validation error close to the best validation

| Instances | depth = 2 | | | | | depth = 3 | | | | | depth = 4 | | | | |
|---|---|---|---|---|---|---|---|---|---|---|---|---|---|---|---|
| | $10^2$ | $10^3$ | $10^4$ | $10^5$ | $10^6$ | $10^2$ | $10^3$ | $10^4$ | $10^5$ | $10^6$ | $10^2$ | $10^3$ | $10^4$ | $10^5$ | $10^6$ |
| 50 features | s | s | s | s | s | s | s | s | s | s | s | s | s | s | m |
| 100 | s | s | s | s | s | s | s | s | s | s | s | s | s | s | m |
| 150 | s | s | s | s | s | s | s | s | s | m | s | s | s | m | m |
| 200 | s | s | s | s | s | s | s | s | s | m | s | s | s | m | - |
| 250 | s | s | s | s | m | s | s | s | s | m | s | m | m | m | - |
| 300 | s | s | s | s | m | s | s | s | m | m | s | m | m | - | - |

Table 2: Approximate magnitudes of runtimes for training ODTs using STreeD on synthetic data. $s$ for (sub)seconds, $m$ for minutes, and dashes for runtimes over two hours.

error. This significantly reduced the number of leaf nodes, but also significantly reduced the out-of-sample accuracy for other experiments where no such overfitting was observed.

For the synthetic linear data, Fig. 11c shows similar results. In both cases, with little noise, unconstrained CART achieves a higher accuracy but with a much larger tree. However, when either type of noise increases, ODT's test accuracy becomes relatively higher.

These results show that ODTs are not more sensitive to noise than greedy trees when ODTs are properly tuned. Without tuning, given a fixed size limit, ODTs can overfit more than greedy trees. However, with proper tuning, ODTs perform better if the data is more noisy.

**Scalability of Optimal Decision Trees**

A final major difference between optimal and greedy approaches is their scalability. The worst-case runtime of dynamic programming ODT methods grows exponentially with the size of the tree, linearly with the number of instances, and exponentially with the number of binary features. This is also in line with the theoretical analysis by Ordyniak & Szeider (2021), who also observe that the ODT problem hardness is dependent on the maximum tree size, the number of unique values per feature, and the maximum number of features in which any pair of instances can differ. Contrasting this with greedy methods whose runtime only grows linearly with the size of the tree, linearly with the number of features, and log-linearly with the number of instances (under mild assumptions), it is clear that greedy methods scale better in runtime.

Since scalability is one of the limitations of ODTs, we test for what problem sizes the use of ODTs is practically feasible. Table 2 provides an overview of runtimes for the optimal method STreeD in terms of seconds, minutes, and hours when trained on synthetically generated data from a random decision tree. We find that training ODTs up to depth four remains practically feasible for datasets up to approximately 250 binary features for 100,000 instances and 150 binary features for one million instances.

## 5 Practical Recommendations

Based on the outcomes of the experiments above, this section summarizes the main practical takeaways and recommendations for the use of and comparison with optimal decision trees.

**Advantages of optimal decision trees**  The main advantages of optimal decision trees are:

1. ODTs can directly optimize the target objective rather than relying on a proxy splitting criterion (such as Gini impurity instead of accuracy, see Open Question 1).

2. ODTs obtain a better trade-off between accuracy and size (Open Question 3), which means they can be used as compact interpretable and yet accurate machine learning models.

3. ODTs can benefit more from more data than greedy decision trees (Open Question 4) and recent DP methods scale well with increasing the number of instances (for a fixed number of possible splits).

4. ODTs are less sensitive to noise than greedy decision trees (Open Question 5).

**Limitations of optimal decision trees**  The main limitation for ODTs is still scalability, specifically with respect to maximum size and number of possible splits (features):

1. It is hard to learn ODTs with a large size limit (e.g., beyond depth five). Therefore, ODTs are best to be used to learn compact yet accurate models.

2. The scalability of learning ODTs (with DP) strongly depends on the number of available features. Therefore, rather than blindly passing a dataset into the learning algorithm, ODTs can benefit from a careful informed preprocessing that selects or constructs few but meaningful features, for example in an iterative process of feature selection, engineering, and interpretable model training by a domain expert (Rudin, 2019).

The advantages and limitations of ODTs together show that ODTs are best used when:

1. you want an interpretable (small) yet accurate model;

2. your data has few but meaningful (preprocessed) features;

3. your data has many samples;

4. your data is noisy;

5. and/or you want to directly optimize an objective for which no good proxy splitting criterion exists.

If these conditions are met, there is clear advantage to the use of ODTs. In other cases, there is a trade-off between runtime, accuracy, and model size. Table 2 shows that ODTs can be found even for datasets up to a million samples, given a low depth limit (which is essential for interpretability as well) and a low number of features. If the number of features is large, one can either choose to fall back on greedy approaches or to preprocess the features and keep only the most important ones to keep runtime low.

**Best practices for training optimal decision trees**  The benefits of ODTs are, of course, dependent on following the best practices for training ODTs (see Recommendations 2). These best practices are supported by, for example, the ODT learning method used in this paper: STreeD (Van der Linden et al., 2023). To learn ODTs, we recommend:

1. Directly learn the target metric (e.g., accuracy, f1-score, etc., see Open Question 1).

2. Use proper hyperparameter tuning for the size of the tree (Open Question 2).

3. If runtime and accuracy are your main concerns, tune the depth. If size and accuracy are your main concerns, tune the number of nodes or the complexity cost (Open Question 2).

**Recommendations for experimental comparisons**  Finally, we reviewed previous comparisons between greedy learning methods with ODTs (see Section 3), and recommend the following for future comparisons (Recommendations 1):

1. Compare ODTs with greedy methods without a size limit if accuracy is the main concern and with the same size limit if interpretability is the main concern.

2. Compare on datasets with a variety of sizes.

3. Learn ODTs beyond small size limits (e.g., beyond maximum depth two).

4. Tune both ODTs and the greedy learning method according to best practices (see Recommendations 2 and 3).

**Limitations** Our empirical study is limited to binary-axis aligned decision trees for binary classification. In the main text the experiments are limited to binary features while optimizing accuracy. In Appendix D and E we observe that similar results hold for numeric features and balanced accuracy respectively. Given the large similarities between binary decision trees and multi-way split decision trees and also between binary classification and multi-class classification, we assume that our results translate to these cases as well. We consider the similarities between oblique (multivariate) decision trees and axis-aligned decision trees small, and therefore we recommend caution in applying our results to the oblique case. All our ODT results in the main text were obtained with a single optimization method: STreeD. However, this choice only impacts the runtime analysis. All other results apply for any other ODT optimization method given the same objective.

## 6  Conclusion

We experimentally evaluated how ODT learning compares to greedy learning approaches. By evaluating previous comparisons, we first developed a set of best practices and the supporting analysis framework for comparing ODT and greedy methods (Recommendations 1). We then applied these best practices to answer five open questions, leading to the following five conclusions.

First, a major advantage of ODTs is that they can optimize the target objective directly, rather than having to optimize a proxy splitting criterion as is the case for greedy top-down induction algorithms. In our experiments we observed that the traditional decision tree objectives, such as Gini impurity and entropy, perform significantly worse in out-of-sample accuracy compared to optimizing training accuracy directly.

Second, we show that hyperparameter tuning of the tree size is essential when training ODTs to (i) improve accuracy, (ii) prevent overfitting, and (iii) reduce the size of the final tree. If only interested in accuracy, the method of tuning the tree size is less important. If however, size or runtime is important, the choice of hyperparameter tuning method matters, e.g., tuning the depth for faster learning or tuning the size or complexity cost for smaller trees. Based on these first two conclusions we suggest best practices for training ODTs (Recommendations 2).

Third, we confirm that ODTs on average outperform greedy trees by 1-2% in out-of-sample accuracy under the same depth limit. However, we also discuss the limitations of this type of comparison and suggest to instead look at the area under the size-error curve as a metric for the accuracy-interpretability trade-off. Our experiments confirm that ODTs indeed obtain a significantly better accuracy-interpretability trade-off than their greedy counterparts. Although unrestricted greedy trees can outperform depth-limited ODTs in accuracy, they can quickly grow so large that they cannot be directly interpreted anymore. Random forests or neural networks already suffice if accuracy is the only concern. However, ODTs are an ideal candidate if interpretability is required, as they achieve a superior accuracy-interpretability trade-off over greedy trees.

Fourth, our results refute the hypothesis that the differences between greedy and optimal trees diminish with more data. Instead, we observe that depth-constrained greedy methods may fail to recover the true tree, even for large datasets, where ODTs succeed. Unconstrained greedy trees, on the other hand, may outperform ODTs in accuracy, but do so while growing trees that are much larger than ODTs.

Fifth, our results also refute the concern that ODTs are more prone to overfit than greedy trees. We observe the opposite, i.e., ODTs are less sensitive to noise than greedy trees.

Together, these results increase our understanding of how to train ODTs and solidify their importance, specifically to obtain small interpretable trees with high accuracy performance (see Section 5).

Future work should investigate how other heuristics, such as coordinate descent (Carreira-Perpinán & Tavallali, 2018; Dunn, 2018) and evolutionary methods (Barros et al., 2011), compare to both top-down induction heuristics and optimal methods. Additionally, this work motivates to develop better metrics that do not only measure accuracy but also consider interpretability.

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

# A  Extended Review of Previous Comparisons

This appendix provides a more detailed review of previous comparisons between optimal and greedy decision tree learning approaches. In Section 3, we only highlighted the work by Murthy & Salzberg (1995), Bertsimas & Dunn (2017), and Zharmagambetov et al. (2021). Here, we discuss all other papers (that we know of) that compare the two methods empirically. As before, we restrict our review to binary axis-aligned trees. First, we summarize the comparisons from papers that proposed ODT methods. Second, we discuss other papers that compare greedy heuristics and optimal methods. Together, this discussion provides the background for the comparison in Table 1.

**Comparisons in ODT papers.**  Papers that propose new ODT methods typically aim to train a decision tree with a given size constraint that achieves the best out-of-sample performance. Nijssen & Fromont (2007; 2010) compare their optimal DL8 algorithm with J48, an implementation of C4.5. When trained on the same discretized data, without a depth limit, but with the same minimum support constraint, DL8 is significantly better for 9 out of 20 datasets and worse for one, while yielding trees that are 1.5 times larger than J48. However, when J48 is trained without the minimum support constraint and with the non-discretized data, J48 outperforms DL8 on out-of-sample accuracy for most datasets.

Verwer & Zhang (2019) compare the optimal MIP methods BinOCT, DTIP (Verwer & Zhang, 2017), and OCT with depth-constrained CART on datasets with a few thousand instances. They report results without hyperparameter tuning and observe that the ODTs are significantly better for depths two and three and slightly better for depth four (Open Question 3).

Lin et al. (2020) propose GOSDT, an ODT method with a sparsity coefficient. They conclude that GOSDT obtains a better accuracy-interpretability trade-off than other methods, including CART (Open Question 3). This is based on an experiment on six small datasets with a coarse binarization applied to both GOSDT and CART. CART is tuned using the maximum depth parameter (from one to six), instead of tuning the complexity-cost as is normally done. They tune GOSDT using complexity-cost tuning without a depth limit.

Demirović et al. (2022) compare their optimal MurTree algorithm and CART on binarized datasets with up to 43500 instances and 1163 binary features. They run MurTree for different depths (from one to four) and number of nodes, and CART for different depths (from one to four), and report the best test accuracy for each method. They too report an average out-of-sample improvement of 1-2% over CART (Open Question 3).

The same trend appears in other papers that propose new ODT methods. The aim is to show ODTs' superior performance under a fixed depth limit. An exception is (Alès et al., 2024), who compare with unconstrained greedy approaches. Results are often shown for fixed hyperparameters (Hu et al., 2020; Mazumder et al., 2022; Liu et al., 2024). Scalability limits the analysis to small datasets (Hu et al., 2020; Günlük et al., 2021; Alès et al., 2024) or larger datasets are run only with a maximum depth of two (Hua et al., 2022).

**Other comparisons.**  Papers that do not propose new ODT methods typically have another aim: the best out-of-sample accuracy without imposing depth constraints on the tree.

Dietterich (1995) concludes from the empirical results by Quinlan & Cameron-Jones (1995) that optimal methods are more prone to overfitting (Open Question 5). Exhaustively searching through all possible models may yield smaller models, but is also more prone to finding small patterns that do not represent the ground truth. Therefore Dietterich concludes that it is better to train greedy methods with a model complexity penalty.

Sullivan et al. (2024) propose MAPTree, a search algorithm that finds the maximum a posteriori tree. They compare with DL8.5, GOSDT, and CART and conclude that MAPTree outperforms these approaches, which leads them to question the 'optimality' of ODTs. They observe that DL8.5 is prone to overfitting, while GOSDT is prone to underfitting, and both are sensitive to hyperparameter tuning, whereas MAPTree is not. However, these results are from averaging the performance per hyperparameter setting over all datasets, rather than tuning the hyperparameters for each individual run. Additionally, they evaluate MAPTree without a depth limit, while other methods (including CART) are run with a depth limit.

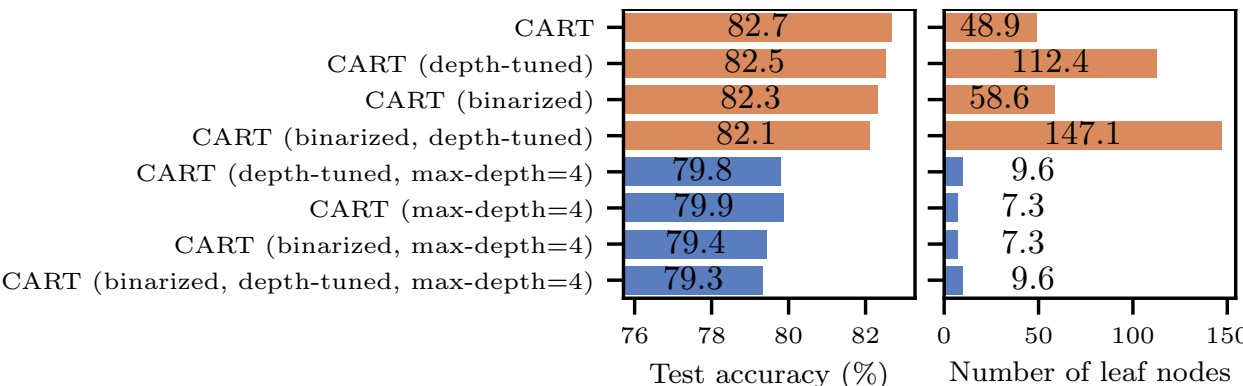

Figure 12: Results for CART (Gini) when trained with(out) binarization, with(out) a depth limit, and when using depth or cost-complexity tuning. Using a depth limit (as indicated in blue) significantly impacts performance. Tuning the depth has a more negative impact for large maximum depths. Binarization with 10 quantile thresholds has no significant impact on the accuracy but does impact the tree size.

Marton et al. (2024) learn axis-aligned trees with gradient descent and compare the results a.o. to CART and DL8.5. GradTree outperforms the other methods for binary classification, while CART performs the best for multi-class classification. The ODT approach DL8.5 performs the second worst in both cases: only the evolutionary approach is worse. They tune each method using random search, except for DL8.5 which they fix to a maximum depth of four. The other methods were typically trained up to depths 7-10.

## B  Training CART

We observe that in the comparisons listed in Section 3 and Appendix A, ODTs are often compared to a modified version of CART, for example, to allow for a direct comparison under similar circumstances. We test the impact of these modifications to assess the validity of these previous comparisons and inform future comparisons. We assess the following typical modifications: (i) tuning the depth instead of the complexity cost; (ii) binarizing the feature data; or (iii) running CART while imposing an additional depth constraint.

We compare CART's performance with these modifications against unmodified CART on the 109 OpenML datasets used before. We approximate the unconstrained CART with a maximum depth of 20. We set the constrained depth limit to four, because of its common use in ODT comparisons. As before, the binarized data has up to ten binary features per continuous feature by using thresholds on ten quantiles or one-hot encoding of categorical features with a maximum of ten categories.

In addition to the depth limit, we apply complexity-cost pruning as done in RPart (Therneau et al., 2023): we train a fully expanded tree and obtain the cost-complexity path with all possible complexity cost values from that tree and use the geometric mean to get the midpoints of those values. We use cross-validation to find the best complexity cost parameter among the midpoints and retrain a tree on the full training data with this parameter.

Unlike RPart, we take the best performing complexity cost parameter, and not the largest complexity cost which performs within one standard error of the best performing one. In our preliminary tests, this resulted in larger trees but better out-of-sample accuracy. In some cases, the cost-complexity path returns a huge number of unique possible complexity cost values. To prevent a long grid search over all these values, we use $k$-means clustering to select 29 representative values. Manually, we always add the possibility that the complexity cost can be one, resulting in at most 30 unique possible values for the complexity cost.

Fig. 12 shows CART's performance under these modifications. The largest differences are between the depth-constrained and the unlimited depth variant. Binarization has only a small impact on the performance (this does not necessarily generalize for more coarse binarizations). When a strict depth limit is imposed, tuning the depth instead of the complexity cost has a small impact but for the unlimited depth case, this significantly

hurts CART's performance. Fig. 12 also shows significantly different tree sizes for CART's modifications. Both binarization and depth tuning result in larger trees. From these results, we can conclude the following best practices:

---

**Recommendations 3** (Training CART).

1. Training CART with a depth limit should be clearly stated.
   *CART trained with a depth limit results in significantly different results than CART without a depth limit.*

2. Tuning the depth of CART instead of the complexity cost should be avoided.
   *Tuning CART's depth rather than the cost-complexity yields larger trees.*

3. Training CART on binarized data should be clearly stated.
   *Depending on the binarization, training on binarized data may or may not significantly harm the performance.*

---

## C   Intuition for why Top-Down Induction Requires Concave Objectives

The reason why TDI methods do not optimize accuracy directly is that an accuracy splitting criterion is often unable to find an improving split in unbalanced data. When splitting a node, TDI methods evaluate all possible splits and choose the split that minimizes the splitting criterion value for the resulting class distributions among the new nodes of each possible split. Fig. 13 visually explains why using accuracy as a splitting criterion is worse at distinguishing improving splits than Gini impurity or entropy.

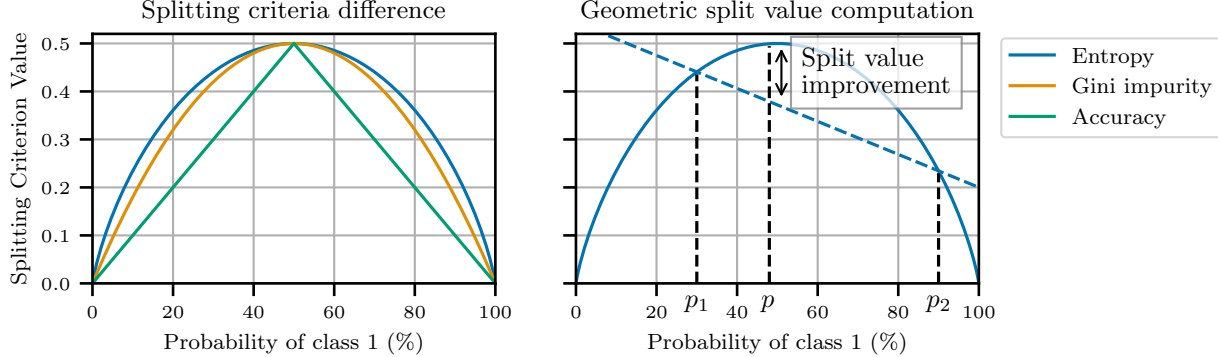

Figure 13: (Left) Three splitting heuristics compared. The horizontal axis shows the binary class distribution expressed as the probability of the first class, and the vertical axis shows the corresponding splitting criterion value (lower is better). (Right) Geometric interpretation of the weighted mean error of two children when $p$, $p_1$, and $p_2$ represent the class distributions of the parent and the two children respectively. The length of the arrow indicates the improvement in the splitting criterion value. Adapted from Flach (2012).

In Fig. 13, the left side shows the function values for accuracy, Gini impurity, and entropy for binary classification. The right side shows how a locally optimal split can be found geometrically. When splitting a node with a probability of the first class of $p$ into two new nodes with probabilities $p_1$ and $p_2$, the new weighted splitting criterion value can be found by drawing a straight line from the criterion value at point $p_1$ to $p_2$. The intersection of the straight line at $p$ is the sum of the weighted criterion value of the two nodes. For Gini impurity and entropy, this value is always lower than the criterion value of the parent node, because both functions are *strictly concave*. Accuracy, however, is not strictly concave, and when $p \leq 0.5, p_1 \leq 0.5$, and $p_2 \leq 0.5$ (or equivalently, all are greater than or equal to 0.5), the weighted sum of the criterion value of the child nodes is the same as that of the parent node. Moreover, for any values $p_1 \leq 0.5$ and $p_2 \leq 0.5$ the weighted sum of the criterion values is the same, and therefore no distinction can be made between

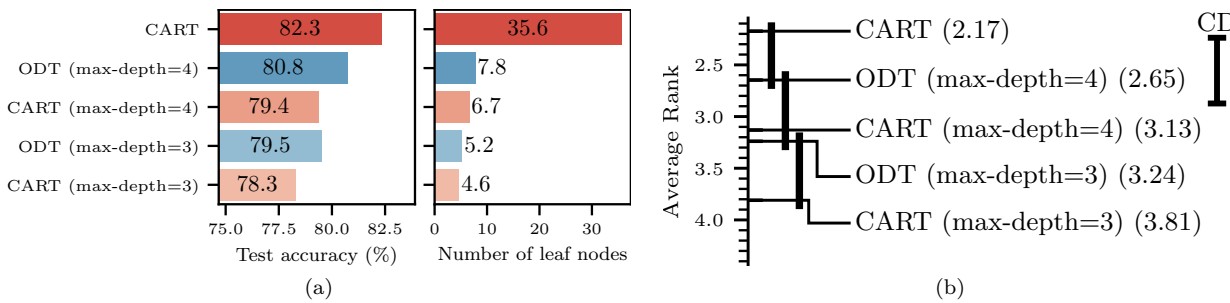

Figure 14: Out-of-sample accuracy of CART and ODT *with the original numeric features* compared on five runs for 92 OpenML datasets. (a) ODT (blue) versus CART (red). (b) Nemenyi critical distance rank test for ODT versus CART.

these splits. Thus TDI heuristics require strictly concave splitting criteria (Kearns & Mansour, 1996) and therefore do not optimize accuracy directly.

## D   Extended Results for Numeric Features

In the main text we report results on datasets with binarized numeric and categorical features to maintain scalability of the ODT approach. Here we report the results for Open Question 3 *without binarizing the numeric features*. We use the state-of-the-art ODT method ConTree (Brita et al., 2025). We use five-fold cross validation to tune the complexity cost. We train ODT and CART trees with maximum depth four, and CART trees without a depth limit. We run ConTree with a 24-hour time limit, and show the results for the 92 datasets that ConTree could solve within this time limit. ConTree timed out on the following datasets: Airlines, Amazon_employee_access, Bank8FM, California, Cardiovascular-Disease-da, Click_prediction_small, Compass, E-CommereShippingData, Electricity, House_8L, Kin8nm, Loan_Status, Mozilla4, Online_shoppers, Pulsar-Dataset-HTRU2, Puma8NH, and Run_or_walk_information.

Fig. 14 shows the results for training CART and ODT with the original numeric features. We again see that the magnitude of the average accuracy difference is approximately the same for depth three and four: 1.2% and 1.4% respectively. However, the Nemenyi critical distance test is not able to confirm that these differences are statistically significant. As in the main text, CART without a depth limit obtains the best out-of-sample accuracy.

## E   Extended Results for Balanced Accuracy

The main text focuses on optimizing accuracy. Therefore, to investigate the impact of optimizing other objectives, we here report the results for Open Question 3 when optimizing for *balanced accuracy*. As also done in the main text, we optimize ODTs using STreeD (Van der Linden et al., 2023) but now with the balanced accuracy objective. We optimize CART trees with class weights set to balanced.

Fig. 15 shows out-of-sample balanced accuracy results for training both CART and ODT with the balanced accuracy objective. For depth three, ODT performs significantly better than CART. For depth four, the average balanced accuracy of ODT is 0.8% higher than CART, but the distance between their average ranks is not large enough to conclude a significant result. For depth-five trees, we see CART get a better average rank, while ODT gets the better average accuracy (on average 0.9% higher). This means there are more datasets where CART gets a few percentage points above ODT, but a few datasets where ODT gets a large difference over CART.

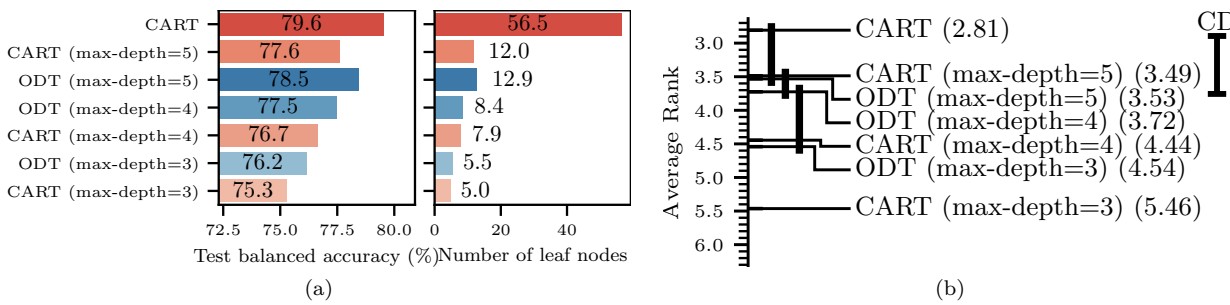

(a)               (b)

Figure 15: Out-of-sample *balanced* accuracy of CART and ODT compared on five runs for all 109 OpenML datasets. (a) ODT (blue) versus CART (red). (b) Nemenyi critical distance rank test for ODT versus CART.

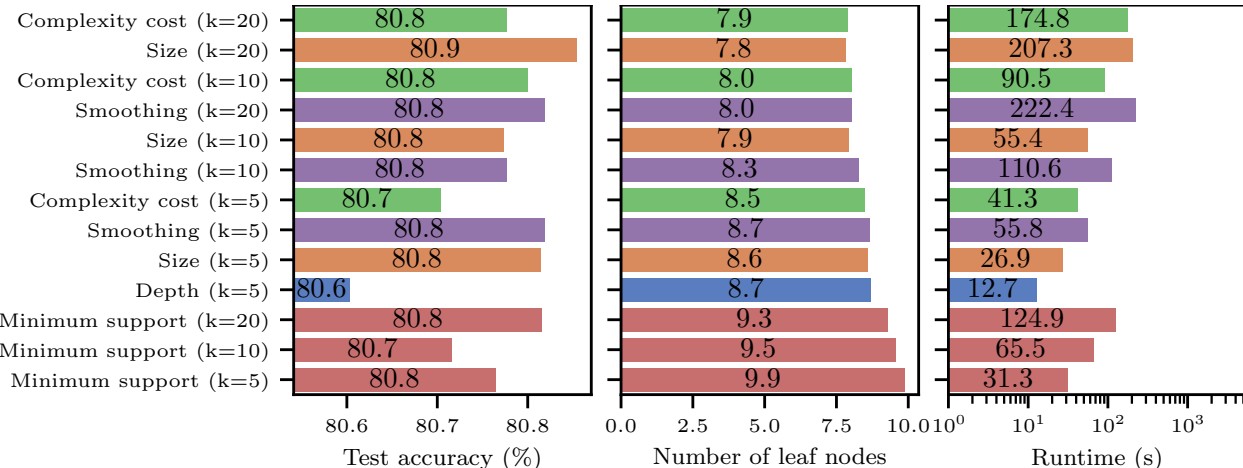

Figure 16: Performance of all tuning methods for $k = 5, 10, 20$ with max-depth $= 4$ on the 108 of the 109 OpenML datasets (Loan_status is left out for exceeding the two-hour time limit). Sorted by the average rank of the number of leaf nodes. The color indicates the tuning method. Test accuracy is roughly the same for all methods. Increasing $k$ typically yields smaller trees, but not better accuracy. As expected, runtime scales roughly linear with $k$.

## F  Extended Results for Hyperparameter Tuning Methods

Here, we report additional experiments on the ODT tuning methods. We test the performance of the methods for other values of $k$, the number of hyperparameter options, and we report the average runtime for each approach. While in the main text, we experimented with max-depth $= 5$ and $k = 10$, here we test for max-depth $= 4$ and $k = 5, 10, 20$.

Fig. 16 shows that the choice of $k$ has no significant impact on the out-of-sample accuracy of the tuning methods. In almost all cases, running with more hyperparameter options results in a lower average number of leaf nodes. As expected, increasing $k$ by a factor two, also increases the runtime for each method by a factor of roughly two.

Fig. 17 shows the results of a Nemenyi critical distance rank test on the *number of leaf nodes* obtained for all datasets. The three tuning methods that obtain on average the smallest trees tune the complexity cost, the size, or the smoothing factor. Tuning the depth or the minimum support leads to significantly larger trees.

The results in Fig. 16 also show that the runtimes of all methods are in the same order of magnitude. The only exception to this is tuning the depth. Therefore, the choice of hyperparameter tuning and the number of parameter options to test mostly depends on how much time one is willing to spend on finding *small* trees and less important for optimizing accuracy.

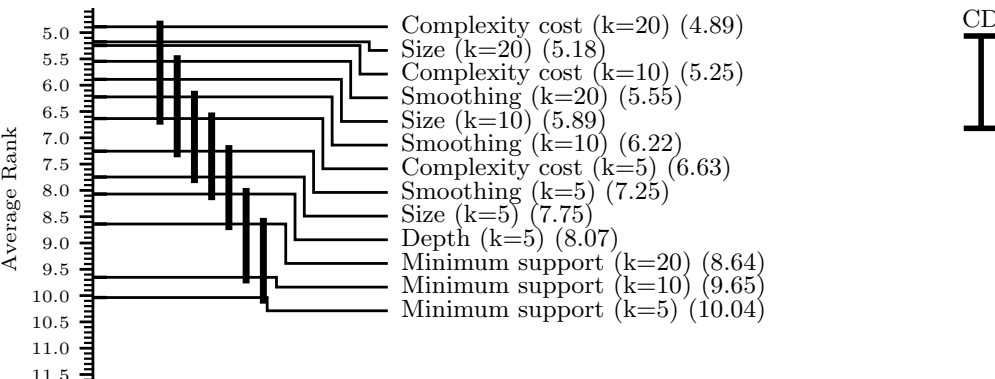

Figure 17: Nemenyi critical rank distance test on the *number of leaf nodes* for each ODT hyperparameter tuning method for the OpenML datasets with maximum depth four. The three methods that obtain the smallest trees tune the complexity cost, size, or the amount of smoothing.

## G  Datasets

Table 3 lists the 109 OpenML datasets used in the experiments in this paper (Vanschoren et al., 2013; Feurer et al., 2021).

| ID | Dataset | Instances | Features | Binarized features | Class imbalance |
|---|---|---|---|---|---|
| 464 | prnn_synth | 250 | 2 | 20 | 0.50 |
| 776 | fri_c0_250_5 | 250 | 5 | 50 | 0.50 |
| 1495 | qualitative-bankruptcy | 250 | 6 | 18 | 0.57 |
| 811 | rmftsa_ctoarrivals | 264 | 2 | 20 | 0.62 |
| 450 | analcatdata_lawsuit | 264 | 4 | 30 | 0.93 |
| 336 | SPECT | 267 | 22 | 22 | 0.79 |
| 1073 | jEdit_4.0_4.2 | 274 | 8 | 66 | 0.51 |
| 23499 | breast-cancer-dropped-mis | 277 | 9 | 37 | 0.71 |
| 1121 | badges2 | 294 | 10 | 43 | 0.71 |
| 40710 | cleve | 303 | 13 | 70 | 0.54 |
| 43 | haberman | 306 | 3 | 26 | 0.74 |
| 1524 | vertebra-column | 310 | 6 | 60 | 0.68 |
| 818 | diggle_table_a2 | 310 | 8 | 79 | 0.53 |
| 1167 | pc1_req | 320 | 8 | 31 | 0.67 |
| 925 | visualizing_galaxy | 323 | 4 | 36 | 0.54 |
| 1011 | ecoli | 336 | 7 | 52 | 0.57 |
| 1048 | jEdit_4.2_4.3 | 369 | 8 | 68 | 0.55 |
| 860 | vinnie | 380 | 2 | 13 | 0.51 |
| 1025 | analcatdata_germangss | 400 | 5 | 16 | 0.78 |
| 909 | chscase_census2 | 400 | 7 | 70 | 0.51 |
| 1511 | wholesale-customers | 440 | 8 | 65 | 0.68 |
| 1498 | sa-heart | 462 | 9 | 79 | 0.65 |
| 814 | chscase_vine2 | 468 | 2 | 18 | 0.55 |
| 724 | analcatdata_vineyard | 468 | 3 | 29 | 0.56 |
| 4329 | thoracic_surgery | 470 | 16 | 54 | 0.85 |
| 767 | analcatdata_apnea1 | 475 | 3 | 20 | 0.87 |
| 884 | fri_c0_500_5 | 500 | 5 | 50 | 0.50 |
| 47049 | Accidents_Prediction_Data | 500 | 6 | 31 | 0.87 |
| 886 | no2 | 500 | 7 | 70 | 0.50 |

| ID | Dataset | Instances | Features | Binarized features | Class imbalance |
|---|---|---|---|---|---|
| 750 | pm10 | 500 | 7 | 70 | 0.51 |
| 40690 | threeOf9 | 512 | 9 | 9 | 0.54 |
| 335 | monks-problems-3 | 554 | 6 | 15 | 0.52 |
| 333 | monks-problems-1 | 556 | 6 | 15 | 0.50 |
| 949 | arsenic-female-bladder | 559 | 4 | 40 | 0.86 |
| 826 | sensory | 576 | 11 | 32 | 0.59 |
| 334 | monks-problems-2 | 601 | 6 | 15 | 0.66 |
| 997 | balance-scale | 625 | 4 | 16 | 0.54 |
| 770 | strikes | 625 | 6 | 60 | 0.50 |
| 827 | disclosure_x_noise | 662 | 3 | 30 | 0.50 |
| 795 | disclosure_x_tampered | 662 | 3 | 30 | 0.51 |
| 774 | disclosure_x_bias | 662 | 3 | 30 | 0.52 |
| 931 | disclosure_z | 662 | 3 | 30 | 0.53 |
| 40981 | Australian | 690 | 14 | 80 | 0.56 |
| 46913 | blood-transfusion-service | 748 | 4 | 34 | 0.76 |
| 46921 | diabetes | 768 | 8 | 73 | 0.65 |
| 1014 | analcatdata_dmft | 797 | 4 | 20 | 0.81 |
| 44268 | anneal | 898 | 38 | 104 | 0.54 |
| 50 | tic-tac-toe | 958 | 9 | 27 | 0.65 |
| 40693 | xd6 | 973 | 9 | 9 | 0.67 |
| 799 | fri_c0_1000_5 | 1000 | 5 | 50 | 0.50 |
| 45604 | dummy | 1000 | 6 | 60 | 0.73 |
| 43255 | 1StudentPerfromance | 1000 | 7 | 43 | 0.52 |
| 46918 | credit-g | 1000 | 20 | 91 | 0.70 |
| 741 | rmftsa_sleepdata | 1024 | 2 | 14 | 0.50 |
| 40702 | solar-flare | 1066 | 10 | 26 | 0.83 |
| 40706 | parity5_plus_5 | 1124 | 10 | 10 | 0.50 |
| 934 | socmob | 1156 | 5 | 31 | 0.78 |
| 40680 | mofn-3-7-10 | 1324 | 10 | 10 | 0.78 |
| 1462 | banknote-authentication | 1372 | 4 | 40 | 0.56 |
| 983 | cmc | 1473 | 9 | 35 | 0.57 |
| 40646 | GAMETES_Epistasis_2-Way_2 | 1600 | 20 | 60 | 0.50 |
| 40649 | GAMETES_Heterogeneity_20a | 1600 | 20 | 58 | 0.50 |
| 46938 | Is-this-a-good-customer | 1723 | 13 | 75 | 0.89 |
| 991 | car | 1728 | 6 | 21 | 0.70 |
| 962 | mfeat-morphological | 2000 | 6 | 43 | 0.90 |
| 914 | balloon | 2001 | 1 | 10 | 0.76 |
| 772 | quake | 2178 | 3 | 30 | 0.56 |
| 40704 | Titanic | 2201 | 3 | 5 | 0.68 |
| 737 | space_ga | 3107 | 6 | 60 | 0.50 |
| 44127 | phoneme | 3172 | 5 | 49 | 0.50 |
| 3 | kr-vs-kp | 3196 | 36 | 38 | 0.52 |
| 871 | pollen | 3848 | 5 | 50 | 0.50 |
| 728 | analcatdata_supreme | 4052 | 7 | 24 | 0.76 |
| 720 | abalone | 4177 | 8 | 73 | 0.50 |
| 46925 | Employee | 4653 | 8 | 33 | 0.66 |
| 40983 | wilt | 4839 | 5 | 50 | 0.95 |
| 45039 | compas-two-years | 4966 | 11 | 34 | 0.50 |
| 44160 | rl | 4970 | 12 | 69 | 0.50 |
| 1460 | banana | 5300 | 2 | 20 | 0.55 |
| 803 | delta_ailerons | 7129 | 5 | 50 | 0.53 |
| 43922 | mushroom | 8124 | 22 | 109 | 0.52 |

| ID | Dataset | Instances | Features | Binarized features | Class imbalance |
|---|---|---|---|---|---|
| 807 | kin8nm | 8192 | 8 | 80 | 0.51 |
| 816 | puma8NH | 8192 | 8 | 80 | 0.50 |
| 725 | bank8FM | 8192 | 8 | 73 | 0.60 |
| 923 | visualizing_soil | 8641 | 4 | 31 | 0.55 |
| 819 | delta_elevators | 9517 | 6 | 47 | 0.50 |
| 46911 | Bank_Customer_Churn | 10000 | 10 | 57 | 0.80 |
| 46924 | E-CommereShippingData | 10999 | 10 | 58 | 0.60 |
| 46948 | PhishingWebsites | 11055 | 30 | 46 | 0.56 |
| 45060 | online_shoppers | 12330 | 17 | 136 | 0.85 |
| 959 | nursery | 12960 | 8 | 26 | 0.67 |
| 1046 | mozilla4 | 15545 | 5 | 40 | 0.67 |
| 44162 | compass | 16644 | 17 | 111 | 0.50 |
| 45558 | Pulsar-Dataset-HTRU2 | 17898 | 8 | 80 | 0.91 |
| 45028 | california | 20634 | 8 | 80 | 0.50 |
| 823 | houses | 20640 | 8 | 80 | 0.57 |
| 43904 | law-school-admission-bian | 20800 | 10 | 53 | 0.68 |
| 843 | house_8L | 22784 | 8 | 75 | 0.70 |
| 45037 | BitcoinHeist_Ransomware | 24780 | 7 | 48 | 0.50 |
| 42493 | airlines | 26969 | 7 | 67 | 0.55 |
| 43900 | amazon_employee_access | 32769 | 9 | 90 | 0.94 |
| 44156 | electricity | 38474 | 8 | 68 | 0.50 |
| 137 | BNG(tic-tac-toe) | 39366 | 9 | 27 | 0.65 |
| 43901 | click_prediction_small | 39926 | 8 | 56 | 0.83 |
| 881 | mv | 40768 | 10 | 75 | 0.60 |
| 46554 | Loan_Status | 45000 | 13 | 94 | 0.78 |
| 45547 | Cardiovascular-Disease-da | 70000 | 11 | 49 | 0.50 |
| 45022 | Diabetes130US | 71090 | 7 | 41 | 0.50 |
| 40922 | Run_or_walk_information | 88588 | 6 | 60 | 0.50 |

Table 3: List of OpenML datasets used in this paper.

Additionally, we perform some experiments on datasets with more than 100,000 instances:

- *covertype* (ID 44121) with 566,602 instances, 10 features, and 100 binarized features.
- *Higgs* (ID 44129) with 940,160 instances, 24 features, and 240 binarized features.

