# OpenReview forum: "Optimal or Greedy Decision Trees? Revisiting their Objectives, Tuning, and Performance"
_TMLR — Under review for TMLR_

### Review · Reviewer_Tv83 · 2026-07-01

**Summary Of Contributions:**

The paper provides an extensive empirical comparison of optimal and greedy decision trees (ODTs and GDTs). ODTs and GDTs differ in terms of the objective function used in training, the former targeting global accuracy directly, rather than local proxies. GDTs avoid the NP-hard training in ODTs through inductive heuristics and are therefore much more widely studied. This empirical study takes advantage of recent developments in scalable ODTs, to advance the conclusions of previous less extensive comparisons.

The conclusions of the investigation are fivefold: 1. that optimising for accuracy is the best choice for ODTs (over globally optimising the proxies used locally in GDT training), 2. that hyperparameter tuning is essential for ODTs, but the specific approach is usually not critical, 3. that prior comparisons with contradictory results can be explained for conditioning on particular depths, but that ODTs have a better trade-off between interpretability (i.e. simplicity of the tree) and accuracy, 4.that gap between performance of GDTs and ODTs grows with more data, rather than shrinking, and 5. that so long as appropriate hyperparameter tuning is performed, ODTs are less likely to overfit than GDTs.

I find the recommendations in Section 3 and the dedicated recommendations section 5 to be well organised and useful for practitioners.

**Audience:**

Yes

**Audience Explanation:**

Yes, I think this would have broad appeal, as decision trees are such widely used interpretable models, and the findings of the paper extend beyond previous comparison studies. Practitioners and students especially are likely to find this a useful resource. That said, it is certainly a research contribution not a pedagogical one.

**Claims And Evidence:**

Yes

**Claims Explanation:**

Yes, the aims of the study are laid out clearly, then addressed systematically. The results for the various research questions appear reproducible and are well evidenced in the paper.

**Requested Changes:**

Questions which may prompt changes in response - not critical but potential to improve clarity

-	What are the implications of the study for multi-class decision trees? Are the recommendations specific to the binary setting?

-	Throughout the synthetic data experiments, the depth of ground truth trees is 3, although the number of features is varied. Is there reason we can expect the conclusions from the depth 3 experiments to carry to other depths?

-	Why is the choice of a maximum n in SE-AUC any less arbitrary than the choice of penalty term in Chaouki et al?


Minor - not critical but not challenging either, I don't think.

•	Abstract line 9, and conclusion line 1 should be ODTs not optimal decision trees.

•	Unify the use of Q1-Q5 or Open Question 1-Open Question 5 throughout the manuscript for clarity, I wasn’t sure on the first(?) use of Open Question on p5 whether that was a reference to this paper or Costa and Pedreira.

•	Add your comparison to Table 1 to highlight that you tick all the columns.

•	At Fig 5 and 7, point back to Fig 3 for guidance interpreting the critical rank distance test plots. Do similar regards to the in-text explanation.

•	P15 *take the average -> takes the average

•	Somewhat confusing to have $n$ as number of nodes in Definition 1 when it is data size elsewhere.

•	State more clearly how the different parameters in step 1 of SE-AUC computation should be chosen – I think the aim is to choose these to realise at least one tree with each number of leaf nodes? How many are needed to get a reasonable estimate?

---

> ### Author Response · Authors · 2026-07-14
> **response to the review**
>
> Thank you very much for the review and for the suggestions for improvement. We have uploaded an updated version of our paper that incorporates the requested changes. Our response to the questions is as follows:
>
> 1. **Multi-class decision trees**
>
> We limited our study to binary classification, considering that this alone is already a large empirical study. We have added to the updated text that we expect our results to carry over to multi-class decision trees.
>
> 2. **Synthetic trees of depth 3**
>
> We also experiment with a linear separator as the ground truth and observe similar results. See Fig. 8c and 11c.
>
> 3. **Arbitrariness of n in the SE-AUC metric**
>
> The sparse objective by Chaouki et al. and others gives a single value for a specific choice of penalty term: the accuracy of a single tree. Our SE-AUC metric, on the other hand, measures the accuracies for a number of trees of different sizes. Additionally, the effect of our parameter n has a diminishing effect since the differences between methods becomes smaller as n increases.
>
> 4. **Editorial comments**
>
> Thank you very much for pointing out the minor issues. We have corrected them.

---

### Review · Reviewer_CQy6 · 2026-07-01

**Summary Of Contributions:**

In this manuscript, the authors conduct a large-scale empirical study comparing optimal decision trees (ODTs) and traditional greedy decision tree learning methods. Since there seems to be a lot of conflicting evidence on the value of ODTs, the authors set out to revisit five open questions concerning ODTs and greedy decision trees - 1) the objectives used for learning each of them, 2) the role of hyperparameter tuning, 3) predictive performance, 4) scalability with increasing data, and 5) susceptibility to overfitting. Using a recently developed scalable dynamic programming framework, they evaluate these questions across 109 real-world datasets and additional synthetic datasets. Their findings suggest that ODTs benefit from directly optimizing the target objective rather than impurity-based proxies, that proper hyperparameter tuning is essential, and that ODTs generally achieve a better accuracy-interpretability trade-off by producing smaller trees with competitive or improved predictive performance. The authors also give practical recommendations for fairly comparing greedy and optimal decision tree methods and provide code to facilitate future benchmarking.

**Audience:**

Yes

**Audience Explanation:**

Yes. I believe this paper would be of interest to particularly researchers working on interpretable machine learning, decision tree learning, empirical evaluation, and optimization-based machine learning.

**Claims And Evidence:**

Yes

**Claims Explanation:**

Yes. Overall, the main claims made by the authors are supported by accurate and reasonably convincing evidence. The paper is primarily empirical, and the authors back their conclusions with a large-scale benchmark over 109 OpenML datasets, additional synthetic experiments, statistical rank-based comparisons, and careful discussion of prior conflicting results.

**Requested Changes:**

I believe the paper is well written and presented however, I would suggest a few changes to mainly help improve the readability of the paper.

1. While the empirical evaluation is extensive, the conclusions are drawn under specific settings (binary classification, axis-aligned trees, the STreeD framework, and primarily binarized features). A clearer discussion of the extent to which these findings generalize to other decision tree variants would strengthen the paper. (Minor)
2. The recommendations are useful, but they could be made even more actionable by including a concise decision guide or summary indicating which type of decision tree method is recommended under different practical scenarios (e.g., dataset size, interpretability requirements, runtime constraints). (Minor)
3. The paper demonstrates the predictive and interpretability benefits of ODTs, but a more explicit discussion of the computational cost versus performance trade-off would help practitioners better understand when ODTs are preferable over greedy approaches. (Major)

---

> ### Author Response · Authors · 2026-07-14
> **response to the review**
>
> Thank you very much for the review and for the suggestions for improvement. We have uploaded an updated version of our paper that incorporates the requested changes. Our response to those requests is as follows:
>
> 1. **Discussion of experiment settings**
>
> We have strengthened our discussion of the results, including the following additions. We consider oblique trees (non axis-aligned) to be a considerably different model type, and therefore we do not assume that our results carry over for this scenario. We expect the results to be similar for multi-class classification. Our results, except runtime, all carry over if another optimization framework than STreeD would have been used. In appendix D we report results for the non-binary case, which has, as expected, no impact on our conclusions.
>
> 2. **Concise decision guide**
>
> We have added a recommendation of when to use optimal or greedy decision trees to our recommendation section.
>
> 3. **Trade-off discussion**
>
> In the aforementioned discussion of when to use which method, we now also discuss the computational cost based on our computation cost analysis in Table 2. Table 2 indicates that for many realistic scenarios we can compute ODTs in the order of seconds or minutes. If the number of features grows too large, an end-user may need to apply feature engineering before training an ODT.

---

### Review · Reviewer_Fia1 · 2026-07-01

**Summary Of Contributions:**

This paper is an empirical comparison of optimal decision trees (ODTs), learned with the dynamic-programming method STreeD, against greedy top-down induction (CART).
It runs on 109 OpenML datasets together with controlled synthetic data.
The improved scalability of STreeD lets the authors reach larger regimes than earlier work, up to $10^5$ instances and depths beyond two, and the study is organized around five contested questions.

Q1 studies the effect of the training objective, and finds that the concave surrogates (Gini, entropy) generalize worse than optimizing accuracy directly when they are optimized globally rather than greedily.
The paper reads strict concavity as a requirement of greedy induction (Kearns and Mansour), not as an intrinsically useful property.
Q2 compares five hyperparameter-tuning schemes for ODTs, and finds that tuning is needed for accuracy while the choice of scheme mainly affects tree size and runtime.
Q3 revisits the ODT-versus-CART accuracy comparison: it isolates the depth-limit condition, confirms the 1 to 2 percent advantage under a fixed depth limit, and restates the advantage as a better accuracy-size trade-off, measured by a proposed area under the size-error curve (SE-AUC).
Q4 and Q5 reject two standing hypotheses, namely that the gap shrinks with more data and that ODTs overfit more than greedy trees.
The paper closes with best practices and an appendix on how common changes to CART (binarization, depth limits, depth-versus-cost tuning) affect the comparison.

The contribution is consolidation and clarification rather than a new algorithm or theorem.
Its value is that it runs the comparison on a common, larger, and more controlled setup, and provides a fair-comparison protocol for later work.

**Audience:**

Yes

**Audience Explanation:**

The direct audience is the decision-tree, interpretable-machine-learning, and tabular-learning community, for which the paper offers a systematic comparison over a conflicting literature and a reusable protocol for future comparisons.
The Q1 result is also relevant to an optimization audience, since it concerns the cost of optimizing a surrogate instead of the target objective, and the difference between local and global optimization of the same criterion.
The audience is concentrated but real.
Under the TMLR criterion the question is interest, not significance, impact, or method novelty, and there is clearly content here that researchers in this area would learn from, so the criterion is satisfied.

**Broader Impact Concerns:**

No.

**Claims And Evidence:**

Yes

**Claims Explanation:**

At the level of the qualitative conclusions, yes.
The design is careful about scope, and the paper reports the cases that do not favor ODTs: unconstrained CART can match or exceed ODT accuracy, the depth limit for parity depends on the dataset, and the Q4 claim of a growing gap holds for tree size but not for accuracy under unconstrained CART.
The CART-training appendix supports the baseline, and the synthetic experiments with ground-truth trees, together with the TDR and FDR diagnostics, give a mechanism for the observed behavior (spurious splits) rather than only endpoint accuracies.
On these grounds the five main answers are convincing in their direction.

The support is weaker for the exact numbers than for the direction, in two respects.
First, the precise magnitudes and the significance verdicts are not yet established, because the sample granularity behind the rank test and the construction of the trade-off metric make the reported numbers optimistic and the significance possibly weaker than claimed.
Second, the Bayesian objective is printed in a form that appears inconsistent with additive minimization, and the MDL expression has a smaller upper-bound inconsistency; therefore parts of the objective comparison need clearer alignment between the formulas and the implementation.
Neither point overturns a qualitative conclusion, and both can be fixed by tightening the claims and the statistical reporting rather than by new experiments.
The specific items are listed under Requested Changes.

**Requested Changes:**

Major concerns:

1. Correct or clarify the Bayesian objective.
As printed, the Bayesian leaf objective is the ratio of Beta functions
\begin{align*}
f_{\mathrm{Bayes}}(n,e) = \frac{B\left(e+\rho_0,\, n-e+\rho_1\right)}{B\left(\rho_0,\rho_1\right)}.
\end{align*}
This is a Beta-Bernoulli marginal likelihood, that is, the Bernoulli likelihood integrated over $\theta$, without the combinatorial factor of the beta-binomial count likelihood.
For a symmetric prior and fixed $n$, the term is largest at a pure leaf (class counts $0$ and $n$) and smallest at a balanced leaf.
Reading $e$ as the misclassification count, so that $e \le \lfloor n/2 \rfloor$ for a leaf that predicts its majority class, the term decreases as $e$ moves from $0$ toward $n/2$, so $f_{\mathrm{Bayes}}$ increases with leaf purity.
Summed over leaves and minimized in the additive DP of Algorithm 2, this quantity would prefer impure leaves, which would invert the intended direction.
Moving to a product formulation would not by itself resolve this, since a MAP tree maximizes the product of leaf marginal likelihoods, equivalently minimizes the sum of their negative logarithms; in that case the additive leaf cost should be $-\log f_{\mathrm{Bayes}}$ up to constants, not $f_{\mathrm{Bayes}}$.
I may be missing a transformation applied inside the solver, so could the authors state exactly which quantity is minimized and print it in that form?
If the implementation already uses the negative-log form, then only the exposition needs to change, but the printed formula should match the code, and the Q1 Bayesian result should be re-checked against the corrected objective.

2. State the sample size and the test behind the critical-distance claims.
I could not tell from the text whether each dataset contributes one averaged rank ($N = 109$) or one rank per fold ($N \approx 545$); could the authors clarify this?
The Nemenyi critical distance scales as $q_\alpha \sqrt{k(k+1)/(6N)}$, hence as $1/\sqrt{N}$, so treating the folds as independent observations would narrow it by about $\sqrt{5} \approx 2.24$ relative to the dataset-level test, which could turn a non-significant gap into a significant one.
Since the folds of a dataset share training data, the standard Demsar protocol uses each dataset as one independent unit and aggregates folds first.
If the analysis already aggregates to the dataset level, then this concern reduces to stating $N$ explicitly; otherwise, it would help to aggregate to one rank per dataset, or to justify the independence assumption, and to report the Friedman omnibus test that the Nemenyi post-hoc presupposes.
Either way, this bears on the significance statements in Q1, Q3, and the appendices.

3. Justify or revise the SE-AUC metric.
As defined, the frontier is traced with test accuracies and then made monotone in size, so the metric appears to report a test-set envelope rather than the performance of a model whose size is selected on validation data.
Because the Pareto correction retains upward test fluctuations, I would expect it to favor a method with a noisier size-accuracy curve, though I am not sure how large this effect is here.
The correction is applied to both methods, so the ODT-over-CART ranking may well hold; my concern is mainly that the absolute values could be optimistic.
Would a variant in which the tree used at each size budget is selected on validation data leave the ranking in Fig. 7 unchanged?
It would also help to report the sensitivity of SE-AUC to the cutoff $n$ and to the uniform weighting over sizes, since a single number can hide the crossovers visible in Fig. 6.

4. Frame the Q1 conclusion around out-of-sample generalization, and control for size.
That direct accuracy optimization gives the best training accuracy is essentially definitional; the non-trivial claim is that it also generalizes at least as well as the concave surrogates, that is, the surrogates do not act as regularizers once the tree is optimized globally.
Please state the conclusion in terms of out-of-sample behavior, so the definitional part is not read as the finding.
As a robustness check, Fig. 3 uses a fixed max-depth without size tuning, and the objectives give slightly different leaf counts (7.5 to 7.9 for ODT, a small spread, and 4.4 to 6.1 for CART), so the accuracy gaps may partly reflect these size differences.
Would a size-controlled version of Fig. 3 at an equal leaf budget, or a validation-based SE-AUC variant, give the same ordering of objectives?
Fig. 7 inherits the SE-AUC issue above, so I would be cautious about presenting it as the clean size-controlled evidence.

Minor concerns:

1. MDL upper-bound inconsistency.
The text first uses the bound $L\left(n, e, \lfloor n/2 \rfloor\right)$ and then defines $f_{\mathrm{MDL}}(n,e) = L\left(n, e, \lfloor (n+1)/2 \rfloor\right)$.
These differ for odd $n$; if they are meant to be the same bound, please use one expression, and otherwise please clarify the difference.

2. Q2 fairness of fixing $k = 10$.
For max-depth $= 5$, depth tuning admits only six distinct values, so the ``same $k$'' budget does not appear to be equalized for the depth method, unless it is handled separately.
Could the authors note how the depth method is treated where the protocol is described?

3. Q4 wording.
``The differences increase rather than decrease'' holds for tree size but not for accuracy under unconstrained CART, where accuracy converges.
The discussion already makes this distinction, so please carry the size-versus-accuracy qualifier into the question-level summary and the abstract, so that the refutation is not overstated.

4. A few editorial items can be fixed in passing: the duplicated phrases "the same approximately the same test accuracy" (Fig. 4 caption), "mostly mostly similar'' (Q5), and "compute compute the accuracy'' (Q3); and the symbol $e$, used for the misclassification count next to $\exp$ and $\ln$, which could be renamed to avoid confusion with Euler's number.

---

> ### Author Response · Authors · 2026-07-14
> **response to the review**
>
> Thank you very much for the review and for the suggestions for improvement. We have uploaded an updated version of our paper that addresses the concerns raised. Our response to the major and minor concerns is as follows:
>
> *Major concerns*
>
> 1. **The Bayesian objective**
>
> In our code we use the sum of the negative logarithms. We have updated the formula in the text accordingly.
>
> 2. **Sample size for statistical tests**
>
> In all cases in our paper, each dataset contributes one averaged rank. We will clarify this in the text.
>
> 3. **SE-AUC metric**
>
> We use the Pareto front of test accuracy rather than the test accuracy for each size because we assume the end user will use cross validation to find the size for which the method performs best. This is, however, not a critical component of the metric, since the conclusion remains the same if we measure the average of test accuracies for each size limit. The SE-AUC cut-off does not significantly affect the conclusion, precisely because we use the Pareto-front, and therefore differences diminish as the cut-off point increases. While a higher cut-off would make the reported effect becomes smaller, our rank test would not be affected by a smaller effect and still make the same conclusion
>
> 4. **Q1 out-of-sample generalization**
>
> We have clarified that the conclusions for Q1 all pertain to out-of-sample generalization. We have also clarified that all results in Fig. 3 are with proper hyperparameter tuning for each method.
>
> *Minor concerns*
>
> 1. **MDL upper bound**
>
> Thanks for pointing out this error. It should be twice $L(n, e, \lfloor(n+1)/2\rfloor)$. We have fixed this in the updated text.
>
> 2. **Fairness of fixing k=10**
>
> For depth, we use 6 settings d=0, … 5. Our results in Appendix F show the results for other values of k and that the conclusions remain the same for these other values.
>
> 3. **Q4 wording**
>
> As noted in the text, we observe differences for both CART unconstrained and CART constrained: in the first case, it is tree size that diverges, in the second, it is accuracy. Therefore, we consider for both cases the original claim to be invalid.
>
> 4. **Editorial items**
>
> Thank you very much for pointing out these issues. We have corrected them.